# Not All Heads Matter: A Head-Level KV Cache Compression Method with Integrated Retrieval and Reasoning

**Yu Fu,**[1] **Zefan Cai,**[2] **Abedelkadir Asi,**[3] **Wayne Xiong,**[3] **Yue Dong,**[1] **Wen Xiao**[3]

[1]University of California, Riverside, [2]University of Wisconsin-Madison, [3]Microsoft

`yfu093@ucr.edu, wxiao@microsoft.com`

## Abstract

Key-Value (KV) caching is a common technique to enhance the computational efficiency of Large Language Models (LLMs), but its memory overhead grows rapidly with input length. Prior work has shown that not all tokens are equally important for text generation, proposing layer-level KV cache compression to selectively retain key information. Recognizing the distinct roles of attention heads in generation, we propose HeadKV, a head-level KV cache compression method, and HeadKV-R2, which leverages a novel contextual reasoning ability estimation for compression. Our approach operates at the level of individual heads, estimating their importance for contextual QA tasks that require both retrieval and reasoning capabilities. Extensive experiments across diverse benchmarks (LongBench, LooGLE), model architectures (e.g., Llama-3-8B-Instruct, Mistral-7B-Instruct), and long-context abilities tests demonstrate that our head-level KV cache compression significantly outperforms strong baselines, particularly in low-resource settings (KV size = 64 & 128). Notably, our method retains just 1.5% of the KV cache while achieving 97% of the performance of the full KV cache on the contextual question answering benchmark. All code and data are available at https://github.com/FYYFU/HeadKV.

## 1 Introduction

Modern Large Language Models (LLMs) increasingly support extremely long inputs: GPT-4 (Achiam et al., 2023), Llama-3 (Dubey et al., 2024), and Qwen-2 (Yang et al., 2024) handle up to 128K tokens, while Claude (Anthropic, 2024) supports up to 1 million tokens. These extended capacities improve performance on tasks like dialogue generation (Li et al., 2024a; Yi et al., 2024), question answering (Ho et al., 2020; Xu et al., 2023), and summarization (Xiao & Carenini, 2019; Koh et al., 2022). As input lengths increase, memory usage and latency grow significantly due to self-attention in transformers. To improve inference speed and efficiency, most LLM inference consists of two phases: prefilling for input processing and decoding for token generation, with key and value states from attention cached for reuse (KV cache). However, as input length increases, KV cache memory grows rapidly, posing significant challenges for storage and efficiency.

To address this, KV cache compression methods (Xiao et al., 2024; Li et al., 2024d; Cai et al., 2024; Feng et al., 2024) have been proposed, typically using token eviction to optimize retention per layer or head during prefilling, reducing memory without impacting performance. However, none have explored varying KV cache size across individual heads. Inspired by prior observations (Voita et al., 2019; Wu et al., 2024; Zheng et al., 2024) that attention heads vary in importance for generation, we propose HeadKV, a **head-level KV cache compression method** that allocates KV cache budgets based on head importance distribution using a novel retrieval and reasoning importance estimation. Specifically, heads deemed more important are allocated larger KV cache budgets, while less important ones receive smaller allocations, optimizing memory usage without sacrificing performance.

In addition to allocating KV cache across attention heads rather than layers, a key aspect of our approach is distributing cache budgets based on head importance measures. Prior work (Wu et al., 2024) proposed method to identify retrieval heads, using importance estimation to assess each head's

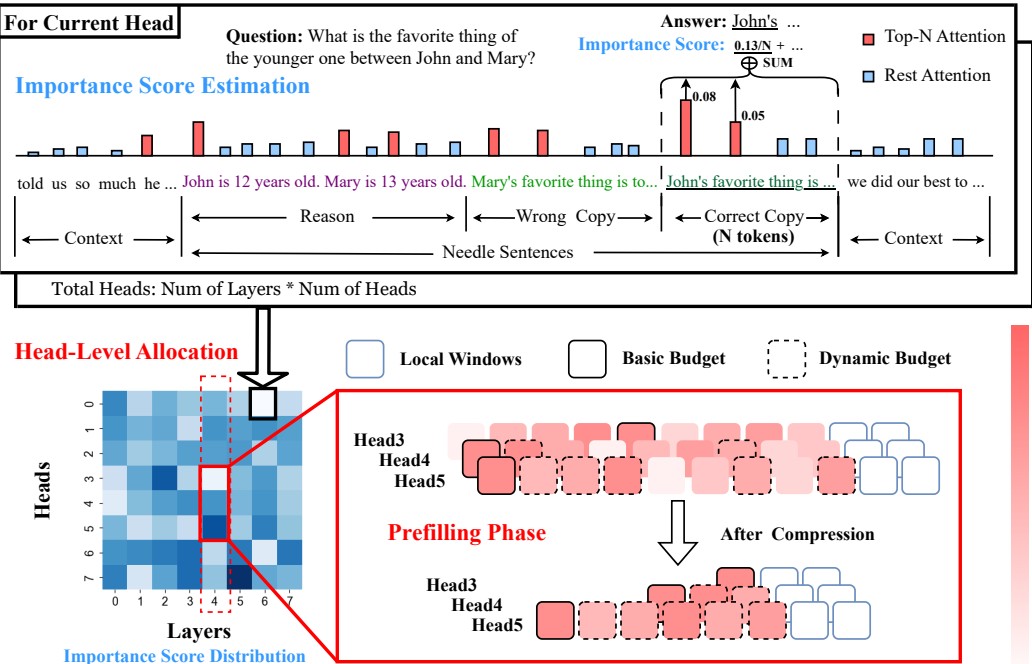

Figure 1: Our proposed head-level KV cache compression method consists of two steps: (1) Head-Level Importance Score Estimation (upper part): important heads that contribute to the **contextual reasoning ability** are identified using Needle-in-a-Haystack tests. (2) Head-Level KV Cache Allocation (lower part): KV cache budgets for each head during the prefilling phase are allocated based on the importance score distribution identified in the first step.

role in retrieving relevant context. We integrate their measure with our head-level KV cache compression as a baseline, observing improved performance over layer-level cache compression.

However, we argue that allocating larger KV cache budgets solely to retrieval heads is insufficient for tasks like contextual Question Answering (QA), which requires both retrieval and reasoning to handle long input contexts effectively. To address this, we propose **an importance-score estimation method that jointly evaluates each head's retrieval and reasoning capabilities for KV cache allocation**. Specifically, as illustrated in the Importance Score Estimation section of Figure 1, we construct questions that require both the retrieval and reasoning abilities of the LLM. For instance, in the provided example, the model must first identify "who is younger between John and Mary" (John) by referencing the context of their ages, and then retrieve John's favorite thing. We then estimate the importance score of each head based on the attention scores generated by the model while answering the question. Using the estimated importance scores, we allocate the KV cache budget for each individual head, meaning that heads demonstrating greater importance in retrieval and reasoning retain a larger portion of the KV cache, as shown in the Head-Level Allocation section of Figure 1. Within each head, we then retain only the most relevant KV cache entries, following the strategy proposed in Li et al. (2024d).

We conduct experiments on various benchmarks requiring both retrieval and reasoning abilities, including QA tasks from LongBench (Bai et al., 2024) and LooGLE (Li et al., 2024b), using backbone models such as Llama-3-8B-Instruct (Dubey et al., 2024) and Mistral-7B-Instruct (Jiang et al., 2023a). Our results demonstrate that the head-level KV cache compression method outperforms previous approaches on nearly all tasks. Furthermore, the allocation strategy based on our estimated importance scores—reflecting both retrieval and reasoning abilities—outperforms allocation strategies based on retrieval ability alone. In challenging scenarios like needle-in-a-Haystack and reasoning-in-a-Haystack tests, our methods effectively preserve the model's retrieval and reasoning capabilities. Finally, experiments on memory and latency reveal that our approach significantly reduces memory usage and decoding latency, comparing with the original full KV.

## 2 RELATED WORK

### 2.1 ATTENTION HEADS

Multi-Head Attention, a fundamental component of Transformer architectures (Vaswani et al., 2017), has been extensively analyzed to understand the roles of individual attention heads (Voita et al., 2019; Olsson et al., 2022; Jin et al., 2024; Wu et al., 2024; Zheng et al., 2024), often with goals such as pruning redundant heads(Shim et al., 2021; Kwon et al., 2022) or enhancing interpretability(Olsson et al., 2022; Jin et al., 2024; Zheng et al., 2024). For example, Voita et al. (2019) observed that only a small subset of heads plays a key role in machine translation, typically managing positional information, syntactic parsing, or focusing on rare words. Similarly, Olsson et al. (2022) identified 'induction heads' that implement a copy-and-paste mechanism, enabling models to replicate previously encountered sequences. Additionally, Zheng et al. (2024) provided a thorough overview of recent efforts to characterize the diverse functions of attention heads, categorizing them into four types: Knowledge Recalling, In-Context Identification, Latent Reasoning, and Expression Preparation. A closely related study by Wu et al. (2024) discovered "retrieval heads" that play a crucial role in knowledge acquisition, using Needle-in-a-Haystack tests. These insights into head-level functionality serve as the foundation for our head-level KV cache compression methods, designed to jointly preserve both retrieval and reasoning capabilities during the compression process.

### 2.2 KV CACHE COMPRESSION

Improving computational efficiency for LLMs, particularly for extremely long inputs, has attracted considerable research interest (Shazeer, 2019; Chen et al., 2023; Dao, 2024; Zhou et al., 2024; Ye et al., 2024), including caching key and value vectors in Multi-Head Attention to improve generation efficiency (Ge et al., 2024a; Xiao et al., 2024; Zhang et al., 2023; Li et al., 2024d; Cai et al., 2024). One challenge with full KV cache approaches is that the memory usage increases linearly with input length, leading to significant memory management challenges. Various KV cache compression techniques have been proposed to reduce memory usage and improve inference efficiency. For example, Xiao et al. (2024) addressed the 'attention sink' issue with StreamingLLM, retaining only the first $k$ tokens KV cache. Zhang et al. (2023) applied the Heavy Hitter Oracle strategy to select key cache entries, while Li et al. (2024d) used the attention scores of the last $\alpha$ tokens to identify relevant entries. Cai et al. (2024) introduced PyramidKV, assigning smaller cache budgets to higher layers based on attention matrix patterns.

While these methods have improved performance and efficiency, they largely depend on rigid layer-level allocation strategies, which may not fully optimize KV cache allocation for downstream tasks. Feng et al. (2024) proposed dynamic head-level allocation using attention scores but still relied on layer-level budgeting. Similarly, Tang et al. (2024) employed Retrieval Heads distribution for head-level allocation but retained a Full KV cache for key heads, limiting true head-level compression. In contrast, our approach allocates KV cache solely based on head-level importance scores, independent of layer constraints, resulting in more effective compression and outperforming baselines without increasing decoding latency or memory usage.

## 3 METHOD

This section presents our proposed head-level KV cache compression method, which consists of three key components: (1) identifying important heads and calculating head-level importance score distributions (3.1), (2) using these distributions to efficiently allocate KV cache budgets across heads (3.2), and (3) determining which Key and Value vectors to retain within each head (3.3).

### 3.1 HEAD-LEVEL IMPORTANCE SCORE ESTIMATION

Accurate budget allocation at the head level requires identifying which heads are most and least important for the given task. By leveraging this importance distribution, we can assign larger KV cache budgets to more critical heads and smaller budgets to less significant ones. To achieve this, we propose a novel importance-score estimation method, inspired by Wu et al. (2024), which allows us to effectively estimate the importance of each attention head for optimal KV cache allocation.

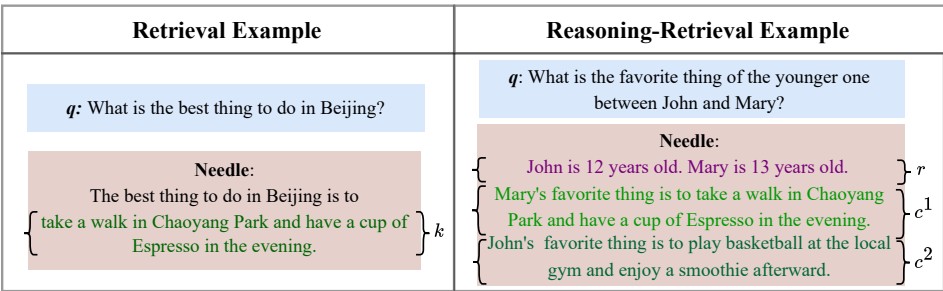

Figure 2: Comparison of examples for head identification: Needle-in-a-Haystack test example from Wu et al. (2024) for identifying Retrieval Heads distribution (left), and our proposed Needle-in-a-Haystack test example for identifying Retrieval-Reasoning Heads distribution (right).

**Retrieval Heads**  Wu et al. (2024) use the Needle-in-a-Haystack test [1] and custom retrieval examples, as shown in Figure 2, to estimate the importance score for each head. In these examples, a question $q$ that cannot be answered using the model's parametric knowledge is paired with an answer $k$ (the "Needle") inserted into a haystack $c$ at different positions $p_i$. The model is required to retrieve the exact answer $k$ from the combined input $(k, c)$.

During each decoding step $t$, a head $h$ earns a fraction of the importance score if (1) the token with the highest attention score $\arg\max(a_h)$ matches the generated token, and (2) the token is part of the inserted answer $k$. The final importance score for each head $h$ is calculated accordingly:

$$S_h = \sum_{t=1}^{N} \mathcal{N}^t, \quad \text{where } \mathcal{N}^t = \begin{cases} \frac{1}{N} & \text{if } \arg max(a_h^t) \in k \\ 0 & \text{otherwise} \end{cases} \tag{1}$$

where $N$ is the length of the inserted answer $k$, and $a_h^t$ is the attention score on the combined input from head $h$ at $t$-sh decoding step. By using various settings of the inserted position $p_i$, they obtain the head-level Retrieval Heads distribution.

Directly using this distribution poses two issues: (1) it focuses on the retrieval-and-paste mechanism, lacking consideration for the contextual and reasoning skills needed for complex questions; and (2) the distribution is too sparse for effective head-level KV cache allocation, with nearly 70% of heads receiving an importance score of zero due to the strict exact-match requirement (Wu et al., 2024).

**Retrieval-Reasoning (R2) Heads**  To address these issues, we propose a new importance score estimation method that accounts for both the retrieval and reasoning abilities of the heads, enabling a more accurate assessment of their significance.

First, we construct retrieval-reasoning examples by adding explicit contextual reasoning steps to the retrieval examples from Wu et al. (2024), as shown in the Retrieval-Reasoning Example part of Figure 2. We further modify the inserted needle into three parts: $k = (r, c^1, c^2)$, where $r$ is the reasoning step, and $c^1$ and $c^2$ are different answers to the refined question $q$. The model must reason with $r$ to retrieve and generate the correct answer $c^2$, avoiding the wrong answer $c^1$.

Secondly, we refine the estimation method by focusing on the entire correct answer $c^2$ (Correct Copy in Figure 1), since all tokens are relevant to the question $q$. This approach aligns with Contextual QA, which requires both retrieval and reasoning abilities. By considering the full correct answer, the importance score for each head $h$ no longer depends solely on the token with the highest attention score. The importance score for head $h$ is calculated as follows:

$$S_h = \sum_{t=1}^{N} \sum_{i=1}^{N} \mathcal{M}_i^t, \quad \text{where } \mathcal{M}_i^t = \begin{cases} \frac{a_i}{N} & \text{if top-}i(a_h^t) \in c^2 \\ 0 & \text{otherwise} \end{cases} \tag{2}$$

---

[1] https://github.com/gkamradt/LLMTest_NeedleInAHaystack

where $a_i \in \boldsymbol{a}_h^t$ represents the $i$-th highest attention score from head $h$, and top-$i(\boldsymbol{a}_h^t)$ is the token with the $i$-th highest score at $t$-th decoding step. We compute the importance score by considering the entire correct answer and increasing the number of tokens evaluated per head (Importance Score Estimation in Figure 1). Intuitively, heads with higher attention on the correct answer $\boldsymbol{k}$ should receive higher importance scores. We further refine the score using attention weights, yielding a more accurate distribution, as shown in Eq. 2.

## 3.2 HEAD-LEVEL KV CACHE ALLOCATION

With the importance scores estimated for each head, we can identify the key heads and allocate the KV cache budget accordingly. In this section, we explain how to incorporate these distributions into head-level KV cache allocation.

**Preliminary** In Multi-Head Attention, for each head $h$ in layer $l$, the embedded input $\boldsymbol{X} = \{x_1, x_2, \ldots, x_n\} \in \mathbb{R}^{n \times d}$ is mapped into different subspaces using the query $\boldsymbol{W}_Q^l$, key $\boldsymbol{W}_K^l$, and value $\boldsymbol{W}_V^l \in \mathbb{R}^{d \times d_h}$ matrices:

$$\boldsymbol{Q}_h^l = \boldsymbol{X}\boldsymbol{W}_Q^l \in \mathbb{R}^{n \times d_h}; \quad \boldsymbol{K}_h^l = \boldsymbol{X}\boldsymbol{W}_K^l \in \mathbb{R}^{n \times d_h}; \quad \boldsymbol{V}_h^l = \boldsymbol{X}\boldsymbol{W}_V^l \in \mathbb{R}^{n \times d_h}. \tag{3}$$

To optimize memory and enhance efficiency, KV cache compression methods (Xiao et al., 2024; Li et al., 2024d; Cai et al., 2024) are employed to discard unimportant KV cache entries while preserving performance. For each head $h$, the compressed KV cache is reduced to $\boldsymbol{K}_h^l \in \mathbb{R}^{s \times d_h}$ and $\boldsymbol{V}_h^l \in \mathbb{R}^{s \times d_h}$, where $s \ll n$, resulting in a significant improvement to computational efficiency.

**Head-level Allocation** Previous works on KV cache compression during the prefill phase (Xiao et al., 2024; Li et al., 2024d; Cai et al., 2024; Feng et al., 2024) are limited to layer-level allocation, using either uniform or dynamic budgets per layer, but treating all heads within a layer equally. While Feng et al. (2024) incorporate head-level information, their approach still depends on layer-level allocation as a prerequisite.

Building on the head-level importance distributions, we propose a comprehensive KV cache allocation strategy. Each head $h$ is initially assigned a fixed KV cache size $b$ with an associated importance score $S_h$. To allow dynamic allocation, we create a shared budget pool $B$ by extracting a portion of the budget from each head, leaving the remainder as the basic budget. This process is illustrated in the Head-Level Allocation section of Figure 1. The budget pool $B$ is then distributed among the heads in proportion to their importance scores $S_h$. The importance score distribution $\boldsymbol{S}$ was L1-normalized to ensure that the sum of $S_h$ equals to 1. The final head-level KV cache allocation is as follows:

$$B = \frac{b}{\beta} \times L \times H; \quad \boldsymbol{b}_h = (b - \frac{b}{\beta}) + S_h \times B \tag{4}$$

where $b$ is the initial fix budget for each head, $\beta$ is a hyper-parameter to control the size of the dynamic budget pool, $L$ and $H$ is the numbers of layers and heads of current LLM respectively.

The last $\alpha$ instruct tokens are preserved before forming the dynamic budget pool $B$ to guide the selection process, as detailed in Section 3.3. The retained KV cache for each head includes: (1) the basic budget $(b - \frac{b}{\beta})$, (2) the dynamic budget $S_h \times B$, proportional to its importance score, and (3) the last $\alpha$ instruct tokens.

## 3.3 KV CACHE SELECTION

After determining the number of KV cache entries to retain using the above algorithm, we apply an attention-based selection strategy from prior works (Li et al., 2024d; Cai et al., 2024; Feng et al., 2024) to keep the most relevant entries. Specifically, the last $\alpha$ instruction tokens (local windows) guide KV cache selection for each head. Attention scores from these local windows to the remaining tokens are aggregated through a pooling layer, with higher-scoring tokens considered more important and retained in the cache.

# 4 EXPERIMENTS AND ANALYSIS

This section outlines the experimental setup, including KV cache baselines and implementation details. We also conduct additional experiments that: (1) emphasize the importance of enhancing contextual reasoning in importance score estimation (4.3); (2) use the Needle-in-a-Haystack and Reasoning-in-a-Haystack tests to demonstrate how our head-level KV cache compression improves long-context retrieval and reasoning (4.4); and (3) provide a comprehensive comparison with previous methods, showing our approach delivers superior performance while maintaining computational efficiency (4.5).

## 4.1 EXPERIMENT SETTINGS

**Models and Datasets** We compare our head-level KV cache compression method against strong baselines using two open-source LLMs: Llama-3-8B-Instruct (Dubey et al., 2024) and Mistral-7B-Instruct (Jiang et al., 2023a). The evaluation is based on two benchmarks for long-context understanding: LongBench (Bai et al., 2024) and LooGLE (Li et al., 2024b). For LongBench, we use datasets from the Single-Doc QA and Multi-Doc QA categories to assess contextual reasoning. For LooGLE, we focus on the Long Dependency QA task, which includes four QA-related tasks. Dataset details are in Appendix Table 5.

**Baselines and Settings** We evaluate three strong KV cache compression methods as baselines, ensuring all retain the same number of KV cache entries for fair comparison:

1) **SnapKV** (Li et al., 2024d) uses the last $\alpha$ tokens as local windows to guide KV cache selection. Attention scores from these windows to the remaining tokens are pooled to cluster information and guide the selection process.

2) **PyramidKV** (Cai et al., 2024) follows a pyramid attention pattern, allocating more KV cache to lower layers to retain key information, while reducing the budget for higher layers where information is already aggregated.

3) **Ada-KV** (Feng et al., 2024) dynamically allocates budgets to heads within each layer based on their concentration degrees, and can be combined with SnapKV or PyramidKV. Ada-SnapKV is used as the baseline due to its superior performance over Ada-PyramidKV.

Our proposed head-level KV cache compression method also requires a strategy to guide KV cache selection after allocating the budget. Therefore, we use the SnapKV method to guide the selection process for each head. We set the size of the local windows $\alpha = 8$ for both the baselines and our method. The hyper-parameter $\beta$, which controls the size of the shared budget pool, was chosen from $\{1.005, 1.01, 1.1, 1.2, 1.5, 2, 5, 10\}$, and we report the best performance.

## 4.2 MAIN RESULTS

Table 1 lists the evaluation results for contextual tasks from the LongBench and LooGLE benchmarks. Our head-level KV cache compression method consistently outperforms strong baselines, especially with 64 and 128 KV cache configurations. In resource-constrained settings, precise KV cache allocation is crucial. Layer-level methods allocate a fixed cache size to all heads within a layer, making it difficult to retain essential information. Ada-SnapKV improves this by allowing dynamic allocation within layers, but still relies on fixed layer-level budgets. In contrast, our head-level strategy allocates dynamically across individual heads, retaining critical information by adjusting the budget based on each head's importance.

We perform head-level allocation using both the standard Retrieval Heads distribution (HeadKV-R) and our Retrieval-Reasoning Heads distribution (HeadKV-R2) for global KV cache allocation. This combination leads to superior performance across benchmarks. Notably, integrating the Retrieval-Reasoning Heads distribution significantly improves results over the standard Retrieval Heads distribution, highlighting the impact of our approach. Our Retrieval-Reasoning distribution even surpasses the Full-KV cache average (32.90), achieving 32.95 with a 1024 KV cache. Overall, both the head-level allocation strategy and Retrieval-Reasoning distribution are key to these performance gains.

| Method | Single-Doc QA | | | Multi-Doc QA | | | Avg. | Long dependency QA | | | | Avg. |
|---|---|---|---|---|---|---|---|---|---|---|---|---|
| | NartvQA | Qasper | MF-en | HotpotQA | 2WikiMQA | Musique | | Doc.QA | Info. Retrieval | Timeline | Computation | |
| Llama-3-8B-Instruct, KV Size = Full | | | | | | | | | | | | |
| FullKV | 25.56 | 32.07 | 39.71 | 43.57 | 35.28 | 21.18 | 32.90 | 8.73 | 11.21 | 0.67 | 7.43 | 7.01 |
| Llama-3-8B-Instruct, KV Size = 128 | | | | | | | | | | | | |
| SnapKV | 22.11 | 15.79 | 31.01 | 41.12 | 29.20 | 19.35 | 26.43 | 8.36 | 9.46 | **0.79** | 6.56 | 6.29 |
| PyramidKV | 22.01 | 17.05 | 31.52 | 39.27 | 28.99 | 18.34 | 26.20 | 8.89 | 9.63 | 0.61 | 6.72 | 6.46 |
| Ada-SnapKV | 22.99 | 19.95 | 34.22 | 42.97 | 30.82 | 20.15 | 28.52 | 9.07 | 10.3 | 0.54 | 6.59 | 6.63 |
| HeadKV-R | **23.49** | 25.39 | 38.15 | 42.45 | 32.84 | 19.95 | 30.38 | 8.87 | 10.35 | 0.78 | **7.52** | 6.88 |
| HeadKV-R2 | 21.80 | **29.19** | **41.89** | **43.73** | **35.01** | **20.40** | **32.00** | **9.60** | **11.13** | 0.67 | 7.22 | **7.16** |
| Llama-3-8B-Instruct, KV Size = 1024 | | | | | | | | | | | | |
| SnapKV | 25.76 | 27.50 | 38.38 | 43.40 | 34.81 | 20.07 | 31.65 | **9.61** | 11.34 | 0.53 | 7.22 | 7.18 |
| PyramidKV | 25.38 | 26.83 | 36.90 | 44.09 | 34.24 | 21.49 | 31.49 | 8.98 | 11.41 | 0.53 | 6.96 | 6.97 |
| Ada-SnapKV | **25.79** | 29.24 | 38.74 | 43.93 | 36.34 | 19.79 | 32.31 | 8.65 | 11.41 | 0.53 | 7.71 | 7.08 |
| HeadKV-R | 24.85 | **30.94** | **39.82** | 43.52 | **36.58** | 20.37 | 32.68 | 9.20 | **11.67** | **0.55** | 7.71 | **7.28** |
| HeadKV-R2 | 24.66 | 30.82 | 39.56 | **43.97** | 36.47 | **22.24** | **32.95** | 9.02 | 11.51 | 0.47 | **7.85** | 7.21 |
| Mistral-7B-Instruct, KV Size = Full | | | | | | | | | | | | |
| FullKV | 26.63 | 32.99 | 49.34 | 42.77 | 27.35 | 18.78 | 32.98 | 12.17 | 15.52 | 0.49 | 10.03 | 9.55 |
| Mistral-7B-Instruct, KV Size = 128 | | | | | | | | | | | | |
| SnapKV | 21.47 | 21.95 | 45.24 | 33.88 | 21.83 | 15.53 | 26.65 | 10.86 | 12.24 | 0.57 | 8.81 | 8.12 |
| PyramidKV | 21.76 | 21.98 | 43.72 | 32.76 | 22.73 | 15.59 | 26.42 | 10.64 | 11.9 | 0.47 | 8.69 | 7.93 |
| Ada-SnapKV | 21.57 | 24.59 | 46.70 | 35.74 | 25.57 | 14.37 | 28.09 | 11.14 | 12.37 | 0.45 | 9.57 | 8.38 |
| HeadKV-R | 23.97 | **29.60** | 48.40 | 39.66 | 26.31 | **18.13** | 31.01 | 11.43 | 13.04 | 0.53 | **10.26** | 8.82 |
| HeadKV-R2 | **25.04** | 27.95 | **48.48** | 41.28 | **27.65** | 18.05 | **31.41** | **11.44** | 13.08 | **0.63** | 10.20 | **8.84** |
| Mistral-7B-Instruct, KV Size = 1024 | | | | | | | | | | | | |
| SnapKV | 25.38 | 30.22 | 49.29 | 41.84 | 26.60 | 18.08 | 31.90 | 11.69 | 13.89 | 0.52 | 10.54 | 9.16 |
| PyramidKV | 24.28 | 30.05 | 49.17 | 40.49 | 26.43 | 18.80 | 31.54 | 11.77 | 14.51 | 0.51 | 10.19 | 9.25 |
| Ada-SnapKV | 24.82 | 31.49 | 48.80 | 41.18 | 27.38 | 18.22 | 31.98 | 11.96 | 13.82 | **0.53** | 9.92 | 9.06 |
| HeadKV-R | **25.87** | 31.44 | 49.55 | **41.95** | 27.09 | **19.88** | 32.63 | **12.21** | 14.17 | 0.50 | **10.58** | 9.37 |
| HeadKV-R2 | 25.64 | **32.54** | **50.49** | 41.80 | **27.88** | 18.89 | **32.87** | 11.94 | **14.93** | 0.50 | 10.49 | **9.47** |

Table 1: Performance comparison on the LongBench and LooGLE benchmarks for Llama-3-8B-Instruct and Mistral-7B-Instruct. Our head-level KV cache compression method outperforms all baselines, especially in low-resource settings (KV size = 128). It even exceeds the FullKV result (32.90) on Llama-3-8B-Instruct (KV size = 1024, 32.95), highlighting the benefits of incorporating contextual reasoning for head selection.

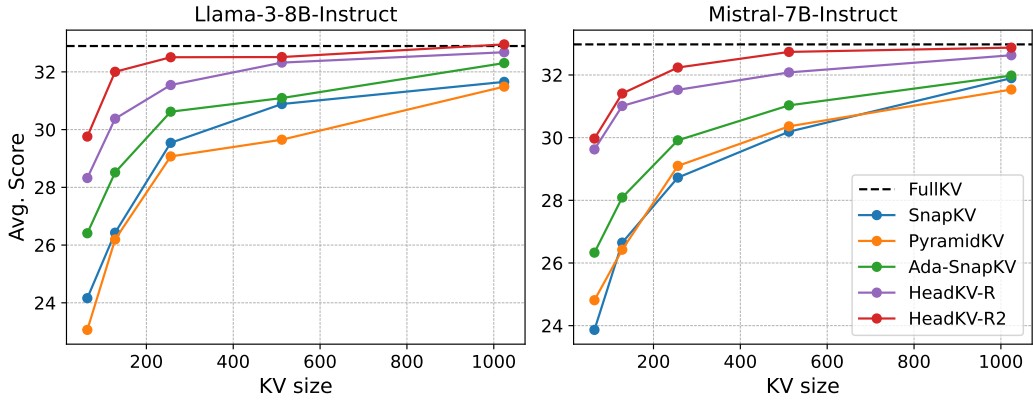

Figure 3: Results for different KV cache sizes (64, 128, 256, 512, 1024), showing average accuracy across six datasets from the LongBench benchmark with an average input length of 8,683 tokens. Notably, a KV cache size of 64 retains just 0.7% of the total tokens.

In addition, we present the results for various retained KV cache sizes (64, 128, 256, 512, 1024) in Figure 3, , with detailed results available in Appendix Table 4.

## 4.3 RETRIEVAL-REASONING HEADS

As detailed in Section 3.1, we propose to improve the standard Retrieval Heads distribution by incorporating retrieval-reasoning examples and refining importance score estimation to better capture

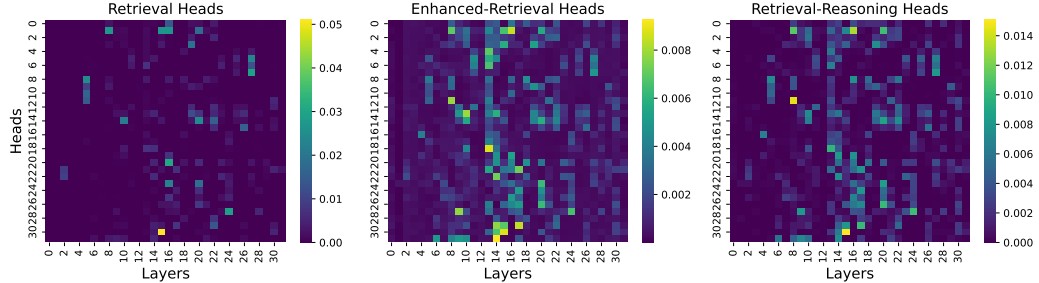

Figure 4: Head visualization for Llama-3-8B-Instruct results. The Retrieval Heads distribution is sparse to effectively differentiate between heads, while our Retrieval-Reasoning Heads has denser distribution for such differentiation. See Appendix Figure7 for Mistral-7B-Instruct results.

| Method | Single-Doc QA | | | Multi-Doc QA | | | Avg. |
|---|---|---|---|---|---|---|---|
| | NartvQA | Qasper | MF-en | HotpotQA | 2WikiMQA | Musique | |
| Llama-3-8B-Instruct, KV Size=128 | | | | | | | |
| HeadKV-R | **23.49** | 25.39 | 38.15 | 42.45 | 32.84 | 19.95 | 30.38 |
| HeadKV-ER | 23.33 | 25.86 | 40.28 | 43.25 | 33.23 | 20.28 | 31.04 |
| HeadKV-R2 | 21.80 | **29.19** | **41.89** | **43.73** | **35.01** | **20.40** | **32.00** |
| Mistral-7B-Instruct, KV Size=128 | | | | | | | |
| HeadKV-R | 23.97 | **29.60** | 48.40 | 39.66 | 26.31 | **18.13** | 31.01 |
| HeadKV-ER | 23.23 | 28.70 | 48.10 | 41.39 | 27.31 | 17.39 | 31.02 |
| HeadKV-R2 | **25.04** | 27.95 | **48.48** | 41.28 | **27.65** | 18.05 | **31.41** |

Table 2: Ablation study results on the LongBench benchmarks. HeadKV-R leverages the standard Retrieval Heads distribution. HeadKV-ER uses the retrieval examples from Wu et al. (2024) but with our proposed importance score estimation method. HeadKV-R2 leverages both our proposed importance score estimation method and the retrieval-reasoning examples.

contextual reasoning and identify relevant heads. We also conduct an ablation study to evaluate the impact of these modifications.

Table 2 presents the ablation study results, while Figure 4 provides visualizations for each distribution. Alongside the standard Retrieval Heads and Retrieval-Reasoning Heads distributions, we introduce the Enhanced-Retrieval Heads distribution, using retrieval examples with our modified importance score estimation method. Comparing Retrieval Heads (HeadKV-R) and Enhanced-Retrieval Heads (HeadKV-ER) reveals that focusing on the entire needle, rather than specific tokens, improves performance. Figure 4 shows that the Retrieval Heads distribution is sparse, while the Enhanced-Retrieval and Retrieval-Reasoning distributions are much denser. The strict constraints on the Retrieval Heads distribution result in most heads receiving a score of zero, leading to a worse results when incorporating Retrieval Heads distributions.

While the Enhanced-Retrieval Heads distribution improves performance slightly, it remains rely on retrieval examples and lacks full consideration of contextual reasoning. In contrast, the Retrieval-Reasoning Heads distribution, reflecting both retrieval and reasoning abilities, consistently outperforms other methods, underscoring the value of incorporating retrieval-reasoning examples.

## 4.4 LONG-CONTEXT RETRIEVAL AND REASONING

We conduct the Needle-in-a-Haystack test to assess the long-context retrieval ability of different KV cache compression methods. As illustrated in Figure 5, we set the KV size to 128 for all methods and keep the other hyperparameters consistent with previous experiments. Results from the Llama-3-8B-Instruct demonstrate that our head-level KV cache compression method effectively retain important information compared with other strong baselines, verifying the superior long-context retrieval ability of our proposed method. However, these tests only retrieve the inserted needle from the haystack and paste it into the generation, lacking an assessment of contextual reasoning ability

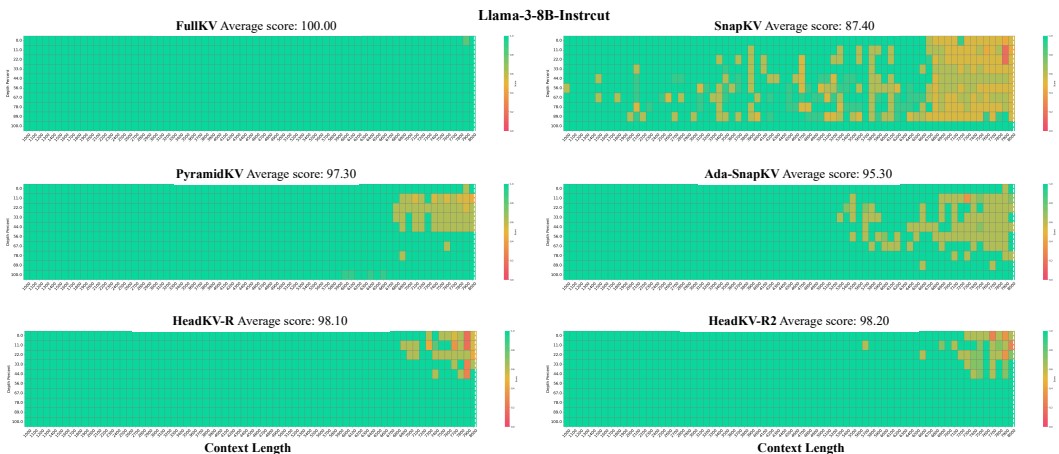

Figure 5: Needle-in-a-Haystack test results on Llama-3-8B-Instruct with KV cache = 128. We build our head-level KV cache method based on SnapKV and our proposed method significantly outperform all strong baselines. Moreover, our Retrieval-Reasoning Heads distribution maintains and improves long context retrieval ability. Results on Mistral-7B-Instruct can be found in Appendix Figure 8, which are consistent with results on Llama-3-8B-Instruct.

| Method | 0k | 1k | 2k | 4k | 8k | Avg. | 0k | 1k | 2k | 4k | 8k | 16k | 32k | Avg. |
|---|---|---|---|---|---|---|---|---|---|---|---|---|---|---|
| FullKV | 63.00 | 59.40 | 55.80 | 56.60 | 50.40 | 57.04 | 61.40 | 55.20 | 52.40 | 40.80 | 40.40 | 35.00 | 31.80 | 45.29 |
| | Llama-3-8B-Instruct, KV Size=128 | | | | | | Mistral-7B-Instruct, KV Size=128 | | | | | | | |
| SnapKV | 60.80 | 57.00 | 54.80 | 52.60 | 45.60 | 54.16 | 56.60 | 50.60 | 46.60 | 34.40 | 35.40 | 33.20 | 29.00 | 40.83 |
| PyramidKV | 61.20 | 58.00 | 52.80 | 52.60 | 47.60 | 54.44 | 58.40 | 50.60 | 46.80 | 36.00 | 36.20 | 31.20 | 28.40 | 41.09 |
| Ada-SnapKV | 61.80 | 59.20 | 53.80 | 53.60 | 46.60 | 55.00 | 58.00 | 51.20 | 46.20 | 35.40 | 36.40 | 31.60 | 28.80 | 41.09 |
| HeadKV-R | 61.60 | 57.00 | 52.60 | 55.00 | 49.60 | 55.16 | 58.40 | 54.00 | 50.80 | **38.00** | 37.40 | 31.80 | **30.20** | 42.94 |
| HeadKV-R2 | **62.60** | **60.40** | **55.00** | **55.80** | 50.40 | 56.84 | **60.20** | 54.20 | 51.20 | 37.20 | **37.40** | **32.60** | 28.80 | **43.09** |

Table 3: Reasoning-in-a-Haystack test results with KV cache = 128. The final results are averaged across QA1-QA5 tasks for each length. Unlike the Needle-in-a-Haystack test, this test inserts multiple needles into the haystack, requiring the model to reason through them to extract the correct answer.

in long-context scenarios. This long-context contextual reasoning ability is crucial for many tasks, such as question answering (QA), summarization (Kuratov et al., 2024; Li et al., 2024c). Therefore, we conduct the Reasoning-in-a-Haystack test to evaluate the long-context reasoning ability of each KV cache compression method across different scenarios.

We follow the setup from Kuratov et al. (2024) to conduct the Reasoning-in-a-Haystack test. This test enhances the bAbI benchmark (Weston et al., 2015), designed for reasoning evaluation, by using text from the PG19 dataset (Rae et al., 2020) as the haystack. Reasoning needles from bAbI are inserted into the haystack, and the model must retrieve and reason through them to generate the correct answer. We use the dataset from Kuratov et al. (2024), averaging results across QA1-QA5 tasks for evaluation. Examples of tasks are shown in Figure 12.

As shown in Table 3, our head-level KV cache compression method significantly outperforms strong baselines, demonstrating its superior long-context reasoning. By incorporating retrieval-reasoning examples, our method achieves even better accuracy, particularly with the Llama-3-8B-Instruct model. Notably, combining head-level KV cache allocation with the standard Retrieval Heads distribution also yields improved results over other baselines. This is due to two factors: first, as shown in Figure 4, there is overlap between Retrieval and Retrieval-Reasoning Heads, indicating heads may serve multiple roles. Second, since the bAbI benchmark contains the answer within the inserted needle (see Figure 12), emphasizing retrieval alone helps our method locate the needle.

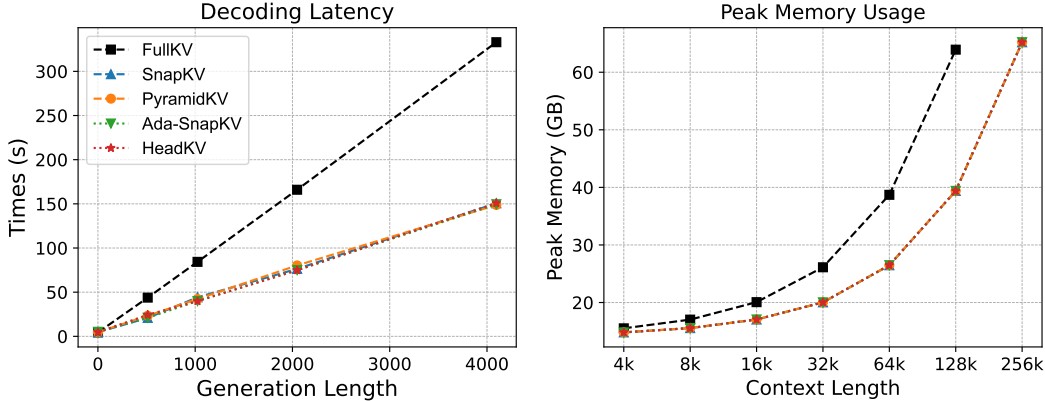

Figure 6: The Decoding Latency and Peak Memory Usage results. Our proposed method maintain the comparable computational efficiency with other KV cache compression baselines.

## 4.5 MEMORY & LATENCY

In this section, we evaluate the computational efficiency of our head-level KV cache compression method using the Mistral-7B-Instruct model with a maximum sequence length of 32K, and FlashAttention (Dao, 2024) as the default setting.

To assess the decoding latency of each method, we use 32K-length data from the Reasoning-in-a-Haystack test as input and set various generation lengths (1, 512, 1024, 2048, 4096) for comparison. As shown in the Decoding Latency of Figure 6, our proposed method achieves the same decoding latency as other KV cache compression methods while maintaining performance closest to the Full KV cache. Notably, the decoding latency includes both the pre-filling time and the decoding time. Therefore, we can conclude that the pre-filling time for our method and other baselines (such as PyramidKV, Ada-SnapKV) is almost negligible, as shown at the starting point (generation length = 1) of Decoding Latency in Figure 6.

In addition to decoding latency, KV cache compression methods also aim to reduce memory usage during the decoding phase. Therefore, we provide the Peak Memory Usage results, as shown in the Peak Memory Usage of Figure 6. All results for Peak Memory Usage are averaged over three trials. Our proposed methods achieve performance comparable to other KV cache compression baselines, significantly reducing memory usage compared to the Full KV cache.

## 5 CONCLUSION AND FUTURE WORK

In this paper, we introduce HeadKV-R2, a head-level KV cache compression method with two key components. We propose a strategy for allocating KV cache budgets across attention heads based on their importance scores and develop a novel approach to estimate these scores by constructing examples that account for both retrieval and reasoning abilities. By allocating larger KV cache budgets to more important heads and smaller budgets to less important ones, our approach efficiently retain important KV cache than layer-level KV cache compression methods. We thoroughly evaluate HeadKV across multiple benchmarks, models, and long-context ability tests, with comprehensive results demonstrating that our method achieves superior performance while maintaining computational efficiency.

For future work, further exploration of various types of attention heads could offer valuable insights, such as those involved in in-context learning (Olsson et al., 2022) and truthfulness (Li et al., 2023). For example, placing greater emphasis on truthfulness heads could help mitigate issues like hallucination, thereby improving the factual accuracy of model outputs. Additionally, it would be worthwhile to investigate the development of a general task-specific score estimation algorithm. One potential direction is to leverage gradients from specific tasks to allocate KV cache budgets more effectively, enabling tailored compression to enhance model performance across diverse applications.

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

# A DETAILE RESULTS

In Table 4, we show the detailed results of Figure 3 in the main paper.

| Method | Single-Doc QA | | | Multi-Doc QA | | | Avg. | Long dependency QA | | | | Avg. | β |
|---|---|---|---|---|---|---|---|---|---|---|---|---|---|
| | NartvQA | Qasper | MF-en | HotpotQA | 2WikiMQA | Musique | | Doc.QA | Info. Retrieval | Timeline | Computation | | |
| Llama-3-8B-Instruct, KV Size=Full | | | | | | | | | | | | | - |
| FKV | 25.56 | 32.07 | 39.71 | 43.57 | 35.28 | 21.18 | 32.90 | 8.73 | 11.21 | 0.67 | 7.43 | 7.01 | - |
| Llama-3-8B-Instruct, KV Size=64 | | | | | | | | | | | | | |
| SKV | 20.51 | 12.80 | 31.69 | 37.02 | 25.91 | 17.02 | 24.16 | 8.84 | 9.43 | **0.66** | 6.18 | 6.28 | - |
| PyramidKV | 21.17 | 13.66 | 29.34 | 34.86 | 23.46 | 15.88 | 23.06 | 8.27 | 9.31 | 0.63 | 6.86 | 6.27 | - |
| Ada-SKV | 22.26 | 17.30 | 33.37 | 39.82 | 27.86 | 17.85 | 26.41 | 9.08 | 9.86 | 0.55 | 6.82 | 6.58 | - |
| Ours. Copy | 22.67 | 23.54 | 37.51 | 37.45 | 29.76 | 19.01 | 28.32 | 8.80 | 10.51 | 0.58 | 6.68 | 6.64 | 2 |
| Ours. Mix | **23.21** | **25.33** | **38.71** | 40.64 | **31.33** | **19.35** | 29.76 | **9.46** | **10.66** | 0.61 | **6.92** | **6.91** | 1.2 |
| Llama-3-8B-Instruct, KV Size=128 | | | | | | | | | | | | | |
| SKV | 22.11 | 15.79 | 31.01 | 41.12 | 29.20 | 19.35 | 26.43 | 8.36 | 9.46 | **0.79** | 6.56 | 6.29 | - |
| PyramidKV | 22.01 | 17.05 | 31.52 | 39.27 | 28.99 | 18.34 | 26.20 | 8.89 | 9.63 | 0.61 | 6.72 | 6.46 | - |
| Ada-SKV | 22.99 | 19.95 | 34.22 | 42.97 | 30.82 | 20.15 | 28.52 | 9.07 | 10.3 | 0.54 | 6.59 | 6.63 | - |
| Ours. Copy | **23.49** | 25.39 | 38.15 | 42.45 | 32.84 | 19.95 | 30.38 | 8.87 | 10.35 | 0.78 | **7.52** | 6.88 | 1.5 |
| Ours. Mix | 21.80 | **29.19** | **41.89** | **43.73** | **35.01** | **20.40** | **32.00** | **9.60** | **11.13** | 0.67 | 7.22 | **7.16** | 1.01 |
| Llama-3-8B-Instruct, KV Size=256 | | | | | | | | | | | | | |
| SKV | 23.38 | 20.18 | 37.65 | 42.80 | 33.23 | 20.01 | 29.54 | 9.04 | 10.59 | 0.53 | 7.53 | 6.92 | - |
| PyramidKV | 23.94 | 20.27 | 36.27 | 42.51 | 31.44 | 19.99 | 29.07 | 8.66 | 10.61 | 0.53 | 6.98 | 6.70 | - |
| Ada-SKV | 24.20 | 24.63 | 37.95 | 43.64 | 33.27 | 20.03 | 30.62 | 9.29 | 11.23 | **0.62** | 7.10 | 7.06 | - |
| Ours. Copy | 23.83 | 29.04 | **39.90** | 42.36 | 33.58 | **20.57** | 31.54 | 9.05 | 11.15 | 0.52 | 7.22 | 6.99 | 1.1 |
| Ours. Mix | **24.68** | **30.49** | 38.59 | **44.32** | **36.41** | 20.54 | **32.51** | 9.47 | **11.56** | 0.54 | **7.65** | **7.31** | 1.1 |
| Llama-3-8B-Instruct, KV Size=512 | | | | | | | | | | | | | |
| SKV | 25.47 | 23.75 | 38.64 | 43.66 | 33.98 | 19.83 | 30.89 | 9.00 | 11.07 | 0.63 | 7.34 | 7.01 | - |
| PyramidKV | 24.69 | 23.65 | 35.10 | 43.25 | 31.16 | 20.06 | 29.65 | 8.90 | 10.62 | **0.74** | 7.57 | 6.96 | - |
| Ada-SKV | **25.73** | 25.44 | 37.84 | 43.78 | 33.95 | 19.83 | 31.09 | 9.22 | 11.18 | 0.53 | 7.42 | 7.09 | - |
| Ours. Copy | 23.84 | 29.21 | **39.79** | 44.41 | 36.09 | 20.59 | 32.32 | 9.13 | **11.61** | 0.56 | 7.12 | 7.11 | 1.2 |
| Ours. Mix | 24.75 | **29.75** | 38.03 | **44.43** | **36.45** | **21.67** | **32.51** | 9.34 | 11.26 | 0.56 | **7.54** | **7.18** | 1.1 |
| Llama-3-8B-Instruct, KV Size=1024 | | | | | | | | | | | | | |
| SKV | 25.76 | 27.50 | 38.38 | 43.40 | 34.81 | 20.07 | 31.65 | **9.61** | 11.34 | **0.53** | 7.22 | 7.18 | - |
| PyramidKV | 25.38 | 26.83 | 36.90 | 44.09 | 34.24 | 21.49 | 31.49 | 8.98 | 11.41 | 0.53 | 6.96 | 6.97 | - |
| Ada-SKV | **25.79** | 29.24 | 38.74 | 43.93 | 36.34 | 19.79 | 32.31 | 8.65 | 11.41 | 0.53 | 7.71 | 7.08 | - |
| Ours. Copy | 24.85 | **30.94** | **39.82** | 43.52 | **36.58** | 20.37 | 32.68 | 9.20 | **11.67** | 0.55 | 7.71 | **7.28** | 1.2 |
| Ours. Mix | 24.66 | 30.82 | 39.56 | **43.97** | 36.47 | **22.24** | **32.95** | 9.02 | 11.51 | 0.47 | **7.85** | 7.21 | 1.2 |
| Mistral-7B-Instruct, KV Size=Full | | | | | | | | | | | | | |
| FKV | 26.63 | 32.99 | 49.34 | 42.77 | 27.35 | 18.78 | 32.98 | 12.17 | 15.52 | 0.49 | 10.03 | 9.55 | - |
| Mistral-7B-Instruct, KV Size=64 | | | | | | | | | | | | | |
| SKV | 19.95 | 18.63 | 38.16 | 31.24 | 21.39 | 13.81 | 23.86 | 10.41 | 11.49 | 0.46 | 9.38 | 7.94 | - |
| PyramidKV | 20.91 | 19.61 | 38.05 | 32.18 | 22.87 | 15.26 | 24.81 | 10.64 | 11.69 | 0.56 | 9.06 | 7.99 | - |
| Ada-SKV | 22.70 | 21.28 | 42.39 | 34.35 | 22.40 | 14.85 | 26.33 | 10.56 | 11.50 | 0.45 | 8.72 | 7.81 | - |
| Ours. Copy | **24.23** | 25.22 | 46.02 | 38.82 | 26.05 | **17.41** | 29.63 | 10.94 | 13.14 | **0.63** | 9.11 | 8.46 | 1.5 |
| Ours. Mix | 21.77 | **26.57** | **48.39** | 40.12 | **26.76** | 16.21 | **29.97** | **11.19** | **13.94** | 0.48 | **9.87** | **8.87** | 1.2 |
| Mistral-7B-Instruct, KV Size=128 | | | | | | | | | | | | | |
| SKV | 21.47 | 21.95 | 45.24 | 33.88 | 21.83 | 15.53 | 26.65 | 10.86 | 12.24 | 0.57 | 8.81 | 8.12 | - |
| PyramidKV | 21.76 | 21.98 | 43.72 | 32.76 | 22.73 | 15.59 | 26.42 | 10.64 | 11.9 | 0.47 | 8.69 | 7.93 | - |
| Ada-SKV | 21.57 | 24.59 | 46.70 | 35.74 | 25.57 | 14.37 | 28.09 | 11.14 | 12.37 | 0.45 | 9.57 | 8.38 | - |
| Ours. Copy | 23.97 | **29.60** | 48.40 | 39.66 | 26.31 | **18.13** | 31.01 | 11.43 | 13.04 | 0.53 | **10.26** | 8.82 | 1.5 |
| Ours. Mix | **25.04** | 27.95 | **48.48** | **41.28** | **27.65** | 18.05 | **31.41** | **11.44** | **13.08** | **0.63** | 10.20 | **8.84** | 1.1 |
| Mistral-7B-Instruct, KV Size=256 | | | | | | | | | | | | | |
| SKV | 22.26 | 24.94 | 48.30 | 36.76 | 25.16 | 14.93 | 28.72 | 11.07 | 12.39 | 0.53 | 9.13 | 8.28 | - |
| PyramidKV | 21.42 | 25.36 | 47.94 | 38.75 | 25.82 | 15.30 | 29.10 | 11.57 | 12.35 | 0.56 | 9.51 | 8.50 | - |
| Ada-SKV | 23.81 | 27.04 | 48.56 | 38.32 | 25.34 | 16.42 | 29.91 | 11.67 | 13.57 | 0.52 | 10.53 | 9.07 | - |
| Ours. Copy | **24.98** | 29.31 | 49.01 | 41.36 | **27.16** | 17.34 | 31.53 | 11.94 | 13.30 | **0.63** | **10.95** | **9.21** | 2 |
| Ours. Mix | 24.94 | **31.02** | **50.76** | **42.11** | 26.14 | **18.47** | **32.24** | **12.37** | **13.88** | 0.48 | 9.86 | 9.15 | 1.005 |
| Mistral-7B-Instruct, KV Size=512 | | | | | | | | | | | | | |
| SKV | 24.18 | 28.87 | 48.74 | 38.84 | 25.48 | 15.04 | 30.19 | 11.96 | 13.47 | 0.52 | 10.50 | 9.11 | - |
| PyramidKV | 23.07 | 28.97 | 48.37 | 39.54 | 25.63 | 16.59 | 30.36 | 11.34 | 13.32 | **0.65** | 10.81 | 9.03 | - |
| Ada-SKV | 24.22 | 29.92 | 48.96 | 40.08 | 25.55 | 17.45 | 31.03 | 12.12 | 14.53 | 0.52 | 10.57 | **9.44** | - |
| Ours. Copy | 24.97 | 30.94 | 49.45 | 42.25 | 26.34 | 18.54 | 32.08 | **12.09** | 13.88 | 0.62 | **10.94** | 9.38 | 1.01 |
| Ours. Mix | **25.59** | **31.33** | 50.26 | 42.66 | **27.20** | **19.37** | **32.74** | 11.62 | **15.61** | 0.50 | 9.97 | 9.43 | 1.005 |
| Mistral-7B-Instruct, KV Size=1024 | | | | | | | | | | | | | |
| SKV | 25.38 | 30.22 | 49.29 | 41.84 | 26.60 | 18.08 | 31.90 | 11.69 | 13.89 | 0.52 | 10.54 | 9.16 | - |
| PyramidKV | 24.28 | 30.05 | 49.17 | 40.49 | 26.43 | 18.80 | 31.54 | 11.77 | 14.51 | 0.51 | 10.19 | 9.25 | - |
| Ada-SKV | 24.82 | 31.49 | 48.80 | 41.18 | 27.38 | 18.22 | 31.98 | 11.96 | 13.82 | **0.53** | 9.92 | 9.06 | - |
| Ours. Copy | **25.87** | 31.44 | 49.55 | **41.95** | 27.09 | **19.88** | 32.63 | **12.21** | 14.17 | 0.50 | **10.58** | 9.37 | 1.5 |
| Ours. Mix | 25.64 | **32.54** | 50.49 | 41.80 | **27.88** | 18.89 | **32.87** | 11.94 | **14.93** | 0.50 | 10.49 | **9.47** | 1.01 |

Table 4: Details results for different KV cache (64, 128, 256, 512, 1024) on Llama-3-8B-Instruct and Mistral-7B-Instruct.

# B DATASET DETAILS

Table 5 provides details of the datasets used in our experiments. To evaluate the retrieval and contextual reasoning abilities of different KV cache compression methods, we use six question-answering

datasets from LongBench (Bai et al., 2024) and four additional question-answering datasets from LooGLE (Li et al., 2024b) as benchmarks. For the LooGLE datasets, we randomly selected 100 examples to form the datasets used in this study.

| Source | Label | Task | Task Type | Eval metric | Avg Len | Language | Nums |
|---|---|---|---|---|---|---|---|
| LongBench | NrtvQA | NarrativeQA | Single-Doc. QA | F1 | 18,409 | EN | 200 |
| LongBench | Qasper | Qasper | Single-Doc. QA | F1 | 3,619 | EN | 200 |
| LongBench | MF-en | MultiFieldQA-en | Single-Doc. QA | F1 | 4,559 | EN | 150 |
| LongBench | HotpotQA | HotpotQA | Multi-Doc. QA | F1 | 9,151 | EN | 200 |
| LongBench | 2WikiMQA | 2WikiMultihopQA | Multi-Doc. QA | F1 | 4,887 | EN | 200 |
| LongBench | Musique | Musique | Multi-Doc. QA | F1 | 11,214 | EN | 200 |
| LooGLE | Doc.QA | Comprehension&reasoning | Long Dependency QA | F1 | 15,498 | EN | 100 |
| LooGLE | Info.Retrieval | Multiple information retrieval | Long Dependency QA | F1 | 14,808 | EN | 100 |
| LooGLE | Timeline | Timeline reorder | Long Dependency QA | F1 | 15,425 | EN | 100 |
| LooGLE | Computation | Computation | Long Dependency QA | F1 | 17,001 | EN | 100 |

Table 5: Details of Datasets.

## C  LONGBENCH RESULTS

We list the evaluation results on the remaining datasets from LongBench (Bai et al., 2024) in Table 6. Our head-level KV cache compression method also outperforms the other baselines, particularly on summarization and Code datasets across Llama-3-8B-Instruct and Mistral-7B-Instruct.

| Method | Summarization | | | Few-Shot Learning | | | Synthetic | | Code | | Avg. |
|---|---|---|---|---|---|---|---|---|---|---|---|
| | GovReport | QMSum | MultiNews | TREC | TriviaQA | SAMSum | PCount | PRe | LCC | RB-P | |
| Llama-3-8B-Instruct, KV Size=Full | | | | | | | | | | | |
| FullKV | 28.71 | 23.26 | 26.64 | 73.5 | 90.48 | 42.33 | 4.80 | 69.25 | 59.29 | 54.05 | 47.23 |
| Llama-3-8B-Instruct, KV Size=128 | | | | | | | | | | | |
| SnapKV | 19.83 | 21.80 | 21.41 | 65.50 | 89.72 | 38.71 | 5.75 | 69.00 | 58.74 | 54.57 | 44.50 |
| Ada-SnapKV | 20.89 | 22.11 | 21.68 | 70.50 | **90.82** | **39.20** | 5.91 | **69.50** | 59.75 | 54.86 | 45.52 |
| HeadKV-R | 21.08 | **22.35** | 22.50 | **71.50** | 89.45 | 38.40 | 5.00 | **69.50** | 60.89 | 59.92 | 46.06 |
| HeadKV-R2 | **21.76** | 22.16 | **23.94** | **71.50** | 90.19 | 38.88 | **6.60** | **69.50** | **61.08** | **60.21** | **46.58** |
| Mistral-7B-Instruct, KV Size=Full | | | | | | | | | | | |
| FullKV | 32.87 | 24.24 | 27.10 | 71.00 | 86.23 | 42.79 | 2.75 | 86.98 | 56.93 | 54.49 | 48.54 |
| Mistral-7B-Instruct, KV Size=128 | | | | | | | | | | | |
| SnapKV | 20.76 | 22.72 | 21.38 | 67.00 | 85.06 | 40.22 | 3.51 | 65.06 | 52.20 | 47.01 | 42.49 |
| Ada-SnapKV | 21.13 | 22.76 | 22.25 | 68.50 | **85.60** | 41.05 | 3.33 | 62.54 | 52.88 | 49.25 | 42.93 |
| HeadKV-R | 22.19 | 22.86 | 22.57 | 69.50 | 85.46 | **41.16** | 3.56 | **74.49** | 54.60 | 50.89 | 44.73 |
| HeadKV-R2 | **24.30** | **23.48** | **24.18** | 70.50 | 85.54 | 40.72 | **4.83** | 72.63 | **55.49** | **51.39** | **45.31** |

Table 6: Results on LongBench (KV cache=128).

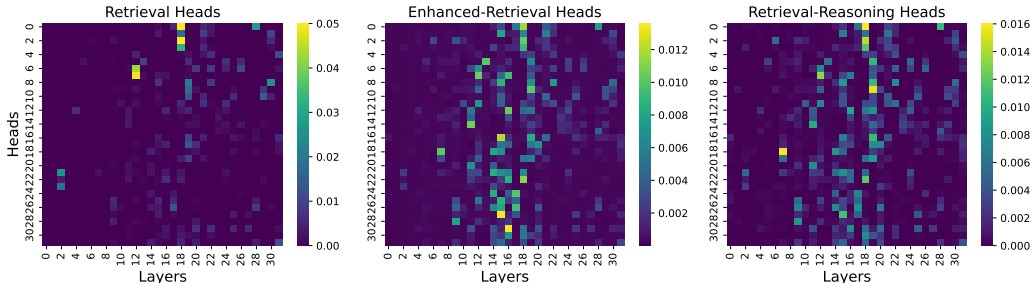

Figure 7: Head visualization results on Mistral-7B-Instruct. Our enhanced distribution can also cover the important retrieval heads.

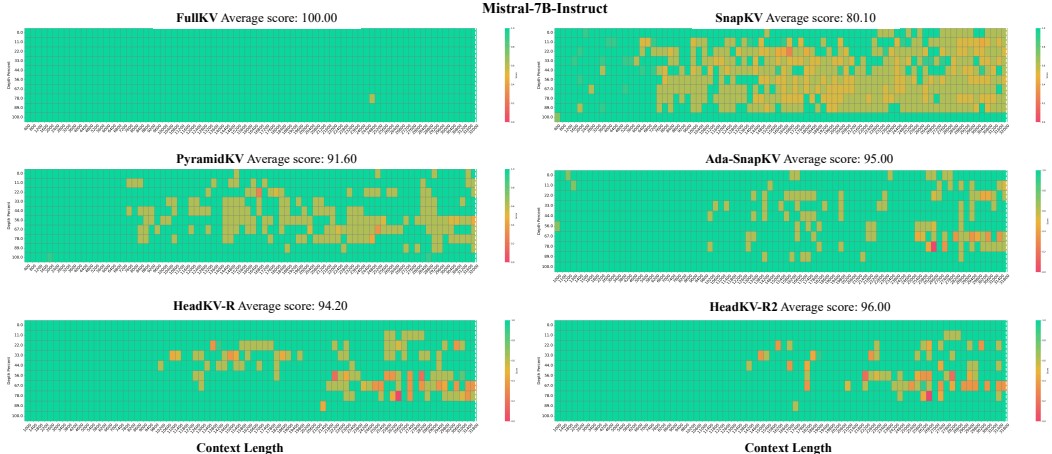

Figure 8: Needle-in-a-Haystack test results on Mistral-7B-Instruct. The results are consist with those on Llama-3-8B-Instruct and our Retrieval-Reasoning Heads distribution (HeadKV-R2 still outperforms other strong baselines.

## D  REASONING-IN-A-HAYSTACK DETAILS

As shown in Figure 12, we provide examples for each task used in the Reasoning-in-a-Haystack test[2]. We use the dataset curated by Kuratov et al. (2024) for the Reason-in-a-Haystack test. Specifically, they use the "Input" listed in Figure 12 as the needle and split it into different sentences using dot. These sentences are then inserted into various positions within the haystack to form the final Reason-in-a-Haystack test examples.

As shown in Figure 13, we provide detailed results across different context lengths and tasks. Our proposed head-level KV cache compression method achieves the best average score among all methods. Notably, on the Llama-3-8B-Instruct model, our method surpasses the Full KV cache on the QA3 task, the most challenging task among the five, as indicated in Kuratov et al. (2024). This suggests that our Retrieval-Reasoning Heads distributions are better at capturing semantically relevant and important content, leading to generating the correct answers.

## E  CONSTRUCT RETRIEVAL-REASONING EXAMPLES

For guiding head-level KV cache compression, we need to obtain the importance score for each head. To achieve this, we manually construct specific examples to ensure that the model relies on heads rather than internal knowledge to answer questions during the Needle-in-a-Haystack experiment. Therefore, we construct retrieval-reasoning examples based on retrieval examples Wu et al. (2024) by introducing different reasoning paths into examples to emphasize the contextual reasoning ability. One constructed Retrieval-Reasoning example is shown in the right of Figure 2. In addition to that example, we reverse the question to create a total of two Retrieval-Reasoning examples.

Following the setup outlined in Wu et al. (2024), in the Needle-in-a-Haystack experiment, we use the model's maximum training length as the maximum haystack length and evenly select 5 different length values as the actual haystack lengths. For each haystack length, the question is inserted at 10 different depths, uniformly distributed from the start to the end of the current haystack length. In total, we generate 100 examples per model to collect Retrieval-Reasoning Head distributions.

The addition of reasoning content $r$ and incorrect answers $c^1$ aims to introduce complexity and context to the reasoning process, which we believe could highlight different heads depending on whether the head supports accurate reasoning patterns or not. Here's how we envisioned their role:

---

[2]https://huggingface.co/datasets/RMT-team/babilong

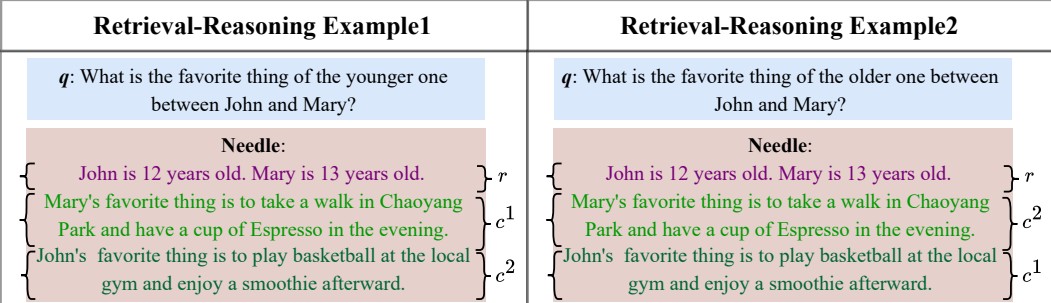

Figure 9: Constructed Retrieval-Reasoning examples. They are used to conduct Needle-in-a-Haystack experiment to obtain the Retrieval-Reasoning Heads distribution.

**Aligning with the requirements of contextual reasoning:** Based on the in-depth analysis on the contextual reasoning dataset, we know that the answer to the corresponding question will still appear in the input but with various distractors. Therefore, the model continues to rely on the retrieval-and-paste mechanism to obtain the true answer. The original retrieval heads estimation method did not account for this phenomenon, but we address it by adding logic and simulating incorrect answers to achieve a more accurate distribution.

**Introducing Diverse Reasoning Paths:** By incorporating both reasoning content and incorrect answers, we are simulating two different potential reasoning paths. The incorrect answer acts as a distractor and we hope to find heads that concentrate on the correct answer even though the incorrect answer has almost the same structure with the correct answer. We expected the correct answer $c^2$ to be treated as the ground truth in the estimation equation, similar to the original Retrieval Heads estimation method.

**Focusing on the correct reasoning path:** The purpose of constructing Retrieval-Reasoning examples is to obtain the importance score for each head, which then guides the head-level KV cache budget allocation. Therefore, our goal is to identify the important heads rather than those focused on incorrect answers. By emphasizing the important heads, the heads that focus on incorrect answers are naturally ignored, as they share the same shared global budget pool.

**Aligning with the standard Retrieval Heads estimation:** we followed the setup in Wu et al. (2024) and determined the Retrieval-Reasoning Heads distributions based on the Needle-in-a-Haystack experiment. Since Needle-in-a-Haystack experiment only outputs the correct answer, we choose to focus on the correct answer to ensure that the maximum importance score each head can achieve in Eq. 2 is 1. Adding additional logic would disrupt this property, potentially affecting the final distribution.

## F  PSEUDO CODE

Codes shown in Listing 1 demonstrates the core steps required for head-level KV cache budget allocation based on the obtained importance score distribution.

- The **obtain_head_budget** function dynamically determines the budget for each head based on the previously obtained importance score distribution as shown in Eq. 4. Since the importance score distribution is static and does not change with the input, this function only needs to be executed once during model initialization.
- The **head_kv** function allocates budgets for different heads. First, we use SnapKV as the selection strategy to guide the specific selection process. After obtaining the budget and determining the selection strategy, we perform the selection and obtain the final retained results based on the budget.

```python
def obtain_head_budget(...):
    # Load saved importance score distribution
    with open(data_path, 'r') as file:
        head_list = json.loads(file.readline())
    # accumulate the importance score and convert to tensor
    head_score_list = [np.mean(l[1]) for l in head_list.items()]
    head_score_list = torch.tensor(head_score_list / sum(head_score_list))
    # Obtain the importance score distribution
    total_attention = head_score_list.reshape(num_hidden_layers, num_attention_heads)
    # Construct shared budget pool and define the minimum KV cache budget
    total_pool_capacity = (base_capacity // beta) * num_hidden_layers * num_attention_heads
    min_num = (base_capacity - base_capacity // self.beta)
    # Head-level Allocation based on the importance score
    head_capacity = torch.round(total_attention * total_pool_capacity + min_num).int()

def head_kv(...):
    # calculate attn_score from the observation windows to input. (SnapKV)
    attn_score = calcul_attn_sore(self, key_states, query_states)
    # Sorted based on attn_score and obtain indices.
    _,indices = attn_score.sort(dim=-1,descending=True)
    # Obtain cached index for each head.
    for head_idx in range(num_heads):
        cache_index = indices[head_idx][...,:head_capacity[self.layer_idx][head_idx]]
        # Expand the indices to match the head dimension for gathering. (Same as SnapKV)
        cache_index = cache_index.view(1, 1, -1, 1).expand(-1, -1, -1, head_dim)
        # Gather the compressed past key and value states based on the selected indices. (Same as SnapKV)
        top_Kcache = origin_heads_key_states[head_idx].gather(dim=2,index=cache_index)
        top_Vcache = origin_heads_value_states[head_idx].gather(dim=2,index=cache_index)
        # Merge with obervation windows
        selected_k = torch.cat([top_Kcache,origin_heads_key_states[head_idx][:, :, -self.window_size:, :]],dim=2)
        selected_v = torch.cat([top_Vcache,origin_heads_value_states[head_idx][:, :, -self.window_size:, :]],dim=2)

        # Combine together
        heads_key_states.append(selected_k.view(-1, head_dim))
        heads_value_states.append(selected_v.view(-1, head_dim))
        # Merge together
    heads_key_states = torch.cat(heads_key_states, dim=0)
    heads_value_states = torch.cat(heads_value_states, dim=0)

    return heads_key_states,heads_value_states
```

Listing 1: Implementation of HeadKV in Pseudo PyTorch style.

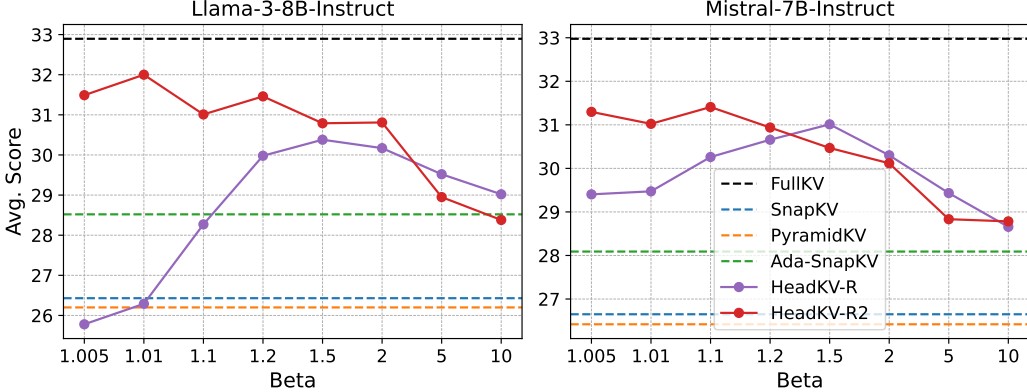

Figure 10: Results for different $\beta$, showing average accuracy across six datasets from the LongBench benchmark.

## G   HYPER-PARAMETER ANALYSIS

The only hyper-parameter introduced by our proposed method is $\beta$, which defines the size of the shared global budget pool $B$. Other hyper-parameters, such as the number of instruction tokens $\alpha$,

are kept consistent with the settings provided in the PyramidKV codebase.[3] We also ensure that all other hyper-parameters are consistent across both the baselines and our proposed method. For the hyper-parameter $\beta$, as we said in Section 4.1, it was chosen from 1.005, 1.01, 1.1, 1.2, 1.5, 2, 5, 10 and we report the best performance according the average score results rather than choosing one $beta$ for each dataset. The details results on Llama-3-8B-Instruct and Mistral-7B-Instruct with KV cache=128 are shown in Figure 11.

For hyper-parameter $\beta$, a smaller value represents a larger shared budget pool $B$ according to Eq. 4, meaning that KV cache allocation relies more heavily on the importance score distribution for allocation. The results show that Head-R2 performs better with a smaller $\beta$, indicating that our Retrieval-Reasoning head distribution is more effective in guiding KV cache budget allocation. It is also worth noting that both Head-R and Head-R2 outperformed Ada-SnapKV, which is current SOTA method, across the different $\beta$ values we set for Llama-3-8B-Instruct and Mistral-7B-Instruct. This demonstrates the stability performance of our proposed head-level KV cache compression method.

## H    DISCUSSION

In addition to KV cache compression methods, another approach to mitigating the long-context scenarios during the pre-filling stage and enhancing the model's ability to handle long texts is context compression (Jiang et al., 2023b; 2024; Ge et al., 2024b; Qin et al., 2024). Within the field of context compression, there are also different directions exploring how to compressing prompts. For example, Jiang et al. (2023b; 2024) proposed defining an additional module to compress the original prompt while ensuring that the meaning and coherence of the prompt remain intact before and after compression. The compressed text will serve as the input for the subsequent LLMs. On the other hand, Ge et al. (2024b) and Qin et al. (2024) adopted a different compression strategy by compressing original prompt into memory slots instead of text. Ge et al. (2024b) proposed ICAE method, which employs an additional trained in-context autoencoder to compress the input into a fixed-length memory slots, which are then used as the input to LLMs. They pretrained and fine-tuned the proposed in-context autoencoder to enable its capacity and generalization. Qin et al. (2024) further utilize the model's internal hidden states as the retained content after compression, with an additional scorer employed for the corresponding selection.

The target of KV cache compression is the KV cache itself, whereas the goal of context compression is the original input or the corresponding representations derived from the input. From the perspective of context compression, current KV cache compression methods can be viewed as compressing the input context using the model's own knowledge, without relying on trainable modules for selection. For instance, the SnapKV Li et al. (2024d) method, which our approach builds upon, uses the last $\alpha$ tokens as the observation window and selectively retained KV cache based on attention from these tokens. Compared to context compression methods like LLMLingua (Jiang et al., 2023b) and ICAE (Ge et al., 2024b), current KV cache compression methods are simpler, as they do not require defining or training additional components. They ar also easier to achieve higher computational efficiency, as these KV cache compression methods avoid relying on external components to obtain compressed inputs and do not require recomputing the key and value matrices.

## I    LIMITATION

Although head-level KV cache compression methods can maintain computational effciency compared to other baselines, they may introduce additional overhead due to the extra effort required to dynamically manage each head especially at in the parallel execution. The synchronization and computational costs associated with head-level operations may reduce decoding speed in multi-GPU environments, leading to a trade-off between performance and efficiency. Hence, an important challenge remains in how to further optimize head-level KV cache compression to mitigate parallel overhead and maintain practical deployment benefits.

---

[3]https://github.com/Zefan-Cai/PyramidKV/

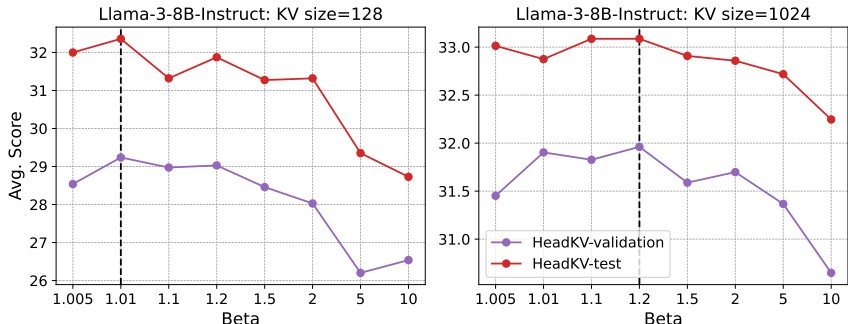

Figure 11: Results for different $\beta$. We leverage scikit-learn to extract 15% of each of the six datasets as a validation set (HeadKV-validation), while the remaining data was used as the test set (HeadKV-test). The black line represents the optimal $\beta$ value determined based on the validation set.

| qa1 | **Input**: John travelled to the hallway. Mary journeyed to the bathroom. Daniel went back to the bathroom. John moved to the bedroom.

**Question**: Where is Mary?
**Target**: bathroom |
|---|---|
| qa2 | **Input**: Daniel took the milk there. John journeyed to the garden. Daniel went back to the hallway. Daniel journeyed to the bathroom. Daniel dropped the milk. Daniel took the milk there. John grabbed the apple there. Sandra journeyed to the kitchen. John went to the hallway. Sandra went back to the garden.

**Question**: Where is the apple?
**Target**: hallway |
| qa3 | **Input**: Mary picked up the apple. John went to the garden. Sandra travelled to the office. Sandra took the milk. John went to the bedroom. Sandra went to the kitchen. John journeyed to the office. Mary left the apple. Mary travelled to the office. Sandra went to the office. Daniel went to the hallway. Sandra discarded the milk.

**Question**: Where was the milk before the office?
**Target**: kitchen |
| qa4 | **Input**: The hallway is east of the bathroom. The bedroom is west of the bathroom.

**Question**: What is the bathroom east of?
**Target**: bedroom |
| qa5 | **Input**: Fred picked up the football there. Fred gave the football to Jeff. Bill went back to the bathroom. Jeff grabbed the milk there. Jeff gave the football to Fred. Fred handed the football to Jeff. Jeff handed the football to Fred. Fred gave the football to Jeff.

**Question**:Who did Fred give the football to?
**Target**: Jeff |

Figure 12: Examples for each task used in Reasoning-in-a-Haystack test.

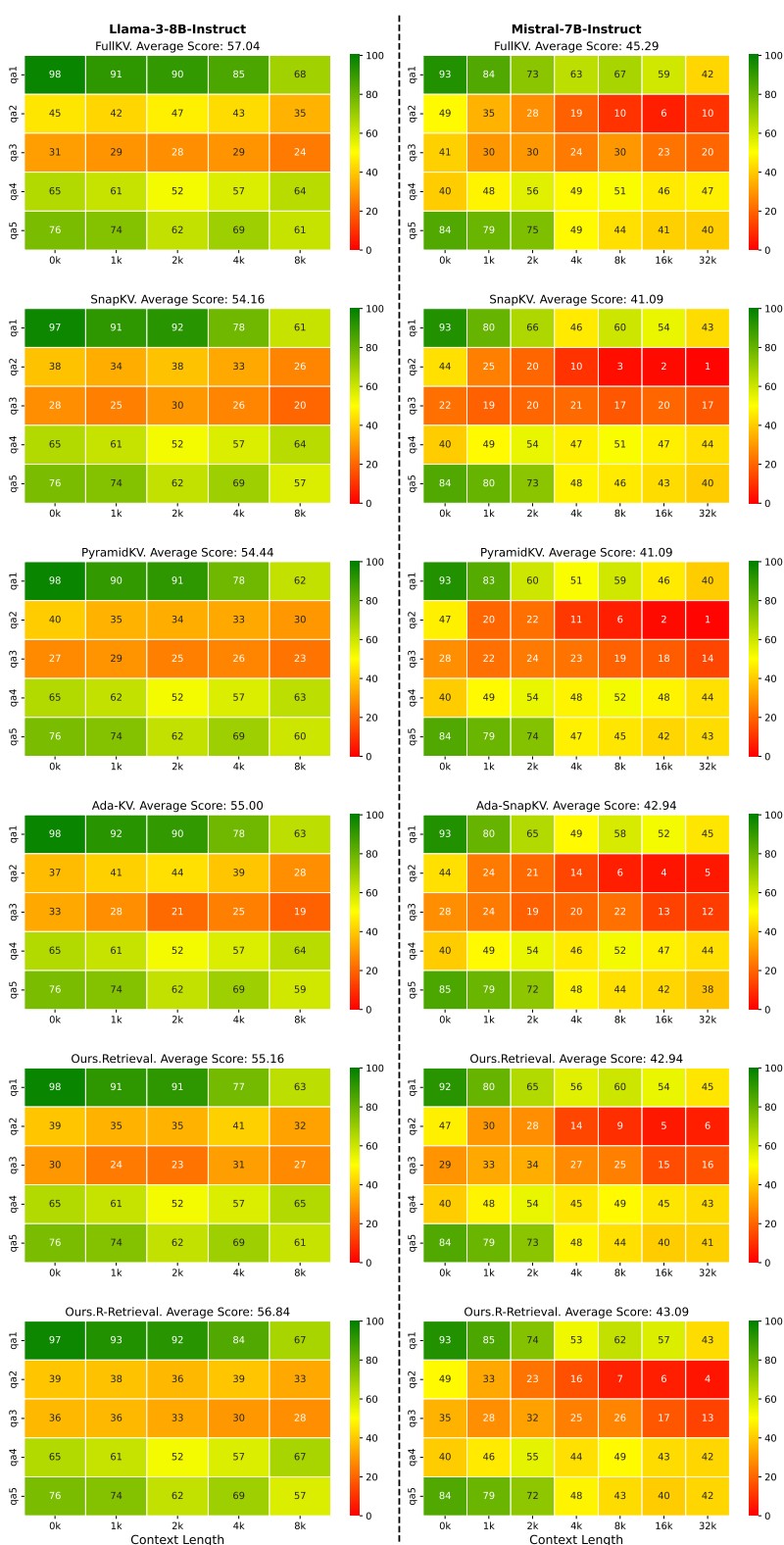

Figure 13: Detail results for Reasoning-in-a-Haystack test on five datasets.

