# OpenReview forum: "Not All Heads Matter: A Head-Level KV Cache Compression Method with Integrated Retrieval and Reasoning"
_ICLR.cc/2025/Conference — ICLR 2025 Poster_

### Official Review · Reviewer_vGF6 · 2024-10-24

**Soundness:** 3
**Presentation:** 4
**Contribution:** 3
**Rating:** 6
**Confidence:** 4

**Summary:**

This paper introduces a method for KV-cache compression for efficient language modeling. While previous work compresses the KV-cache at the global or layer level, this paper leverages the properties of multi-head attention and operates at individual heads, achieving a finer-grained computation allocation. The method weighs the importance of each head by considering their contribution to both the retrieval and reasoning processes, resulting in a better selection of past KV cache. With all strengths combined, the method outperforms the baseline methods in multiple tasks that require long-context retrieval or reasoning.

**post review**: I raised my score from 5 to 6.

**Strengths:**

1. This paper refines the method for head importance scoring proposed by Wu et al. (2024) and the authors justify their refinement with ablation studies.
2. This paper proposes a head-level allocation schema that can dynamically allocate memory across attention heads and layers.
3. The performance of this method is better than baselines under the same memory budget.

**Weaknesses:**

1) The contribution of this paper is relatively marginal in terms of methodology. The modification to the method of Wu et al. (2024) is a shift from retrieval to reasoning and a relaxation of the weighting process, while the overall design of the scoring remains the same. From the KV-cache compression perspective, it improves the granularity of prior work from layer to head without changing the overall compression logic.

2) The application of the KV cache compression is limited to scenarios where retrieval is required. The compression/allocation is already done after reading the instruction part of the prompt, thus the attention of the generation will be constrained to the selection caches. For tasks such as Needle-in-the-Haystack or retrieval-based QA, this strategy might work, but for the rest of the tasks such as summarization or creative generation, it might fail.

This might be a general issue for all the work of KV cache selection, but this paper does not break this limitation.

3) This work might not be practically useful in terms of efficiency. Though the authors claim that their method can be as efficient as other methods and better than the full-KV baseline in section 4.5, the analysis is based on a 7B model. For larger models, the overhead on the KV cache and attention matrix is negligible.

Even for a 7B model, the analysis conducted in this paper shows that the decoding speed is improved only when the decoding sequence length reaches 1000 tokens. For memory usage, the difference is observable when the context length reaches 32k tokens. This might not be a typical case even for the datasets tested in this paper.

**Questions:**

1. What is the experiment condition for the memory usage diagram in Fig 6? The x-axis here is named "context length", so what is the generation length? Also, given the pre-filling phase has not changed, can the proposed method improve the memory usage for context encoding?
2. Finer-grained memory allocation usually means more fragmented memories. From my understanding, the method prunes caches from the token, layer, and head dimensions, will this result in fragmented KV caches? Will this affect the efficiency of parallel computation?
3. It would be great if the authors could discuss and compare this work to RAG methods.

It would be great if the authors discuss other efficiency-concerned papers. A few past work discusses the general compression of KV caches without considering the retrieval process:

- In-context Autoencoder for Context Compression in a Large Language Model. Ge et al, 2024. In ICLR.
- Dodo: Dynamic Contextual Compression for Decoder-only LMs. Qin et al, 2024. In ACL.
- VCC: Scaling Transformers to 128K Tokens or More by Prioritizing Important Tokens. Zeng et al., 2023. In NeurIPS.

---

> ### Author Response · Authors · 2024-11-21
> **Response to Reviewer vGF6 [1/3]**
>
> First, we want to emphasize that long-context scenarios, along with KV cache compression methods designed for these scenarios, are critically important as they lay the foundation for improving computational efficiency. These approaches are essential not only for handling extensive input contexts but also for facilitating seamless transitions to long-generation tasks. By optimizing the management and compression of KV caches during the pre-filling stage, we ensure that the LLMs remain capable of processing large volumes of information while maintaining high-quality and coherent outputs, which is crucial for the next long-generation applications, such as RAG, summarization and long-form QA.
>
> **Q1: Limit Novelty**
>
> Our method mainly consists of two novelty parts: (1) refining the importance score estimation and constructing new retrieval-reasoning examples to obtain the Retrieval-Reasoning Heads distribution, and (2) performing head-level KV cache budget allocation. While Retrieval Heads already exist, effectively combining Retrieval Heads with KV cache compression remains an interesting and worthwhile problem to explore.
>
> + *Importance Score Estimation:* For importance score estimation, we made modifications to the calculation method and incorporated attention scores into the process, as shown in Eq. 2. We compare the results of directly using the standard Retrieval Heads distribution with those obtained using our Retrieval-Reasoning Heads distribution (as shown in Table 1 and Figure 3). The results demonstrate that our refined importance score estimation significantly improves the model's performance across various KV size settings. Compared to the standard Retrieval Heads distribution, our modified estimating approach yields a denser and more precise distribution, enabling better guidance on the amount of KV cache retained for each head (as shown in Figure 4). The results shown in Table 2 further highlight the importance of constructing Retrieval-Reasoning examples rather than relying on standard Retrieval examples.
>
> + *Head-Level KV cache budget allocation:* Next, we further explore how to better utilize the head-level importance score distribution for KV cache compression. Our method is the first to perform fully head-level KV cache budget allocation and achieves SOTA results on Longbench. Other methods, such as RazorAttention, have also recognized the existence of Retrieval Heads distributions. However, they merely use retrieval heads to decide whether to apply FullKV or StreamingLLM to retain KV cache, which leads to limited performance gains. Additionally, such methods often dramatically increase the total number of retained KV cache entries to achieve fair results. In contrast, our approach effectively balances KV cache usage and performance.
>
> Overall, achieving a better importance score distribution and further integrating it with KV cache compression algorithms remains an interesting and under exploration problem. Our efforts on these two aspects provide a potential solution and achieve SOTA performance on the corresponding benchmarks.
>
> **Q2: Limited to retrieval-based scenarios**
>
> In this paper, we focus on KV cache compression during the pre-filling phase to deal with long-context input scenarios.  It is necessary to design different strategies during the pre-filling phase to selectively retain portions of the KV cache, thereby achieving a balance between computational efficiency and performance. The selected KV cache does not simply store information from specific token positions but retains a holistic representation of the overall input. Therefore, for summarization tasks, our proposed method also achieves better results as shown below:
>
> |   |   |   |   |
> |---|---|---|---|
> |Method|GovReport|QMSum|MultiNews|
> |FullKV|28.71|23.26|26.64|
> |SnapKV|19.83|21.80|21.41|
> |Ada-SnapKV|20.89|22.11|21.68|
> |HeadKV-R|21.08|22.35|22.50|
> |HeadKV-R2|21.76|22.16|23.94|
>
>
>
> This table is derived from Table 6 in our paper, where we present the results on Longbench, demonstrating that our proposed method achieves the best overall performance.
>
> For tasks like creative generation, they rely more on the model's inherent capabilities rather than information from the input to generate the corresponding results, which does not align with the objective of this paper. In fact, generating long text is another interesting and worthwhile direction to explore, with relevant works such as [1][2].
>
> [1] Large Language Models Still Exhibit Bias in Long Text
>
> [2] Language Models can Self-Lengthen to Generate Long Texts

---

> ### Author Response · Authors · 2024-11-21
> **Response to Reviewer vGF6 [2/3]**
>
> **Q3: the overhead for larger models**
>
> The following code demonstrates the additional steps required for head-level KV cache budget allocation based on the obtained importance score distribution:
>
> ```python
> # Load saved importance score distribution
> path = `path to the saved distribution file`
> with open(path, 'r') as file:
> 	head_list = json.loads(file.readline())
>
> # accumulate the importance score and convert to tensor
> head_score_list = [np.mean(l[1]) for l in head_list.items()]
> head_score_list = torch.tensor(head_score_list / sum(head_score_list))
>
> # accumulate the importance score and convert to tensor
> head_score_list = torch.tensor(head_score_list / sum(head_score_list))
>
> # Obtain the importance score distribution
> total_attention = head_score_list.reshape(num_hidden_layers, num_attention_heads)
>
> # Construct shared budget pool and define the minimum KV cache budget
> total_pool_capacity = (base_capacity // beta) * num_hidden_layers * num_attention_heads
> min_num = (base_capacity - base_capacity // self.beta)
>
> # Head-level Allocation based on the importance score
> head_capacity = torch.round(total_attention * total_pool_capacity + min_num).int()
> ```
> Ideally, we only need to initialize the required `head_capacity` during the model's initialization phase, since our importance score distribution is static, and we do not need to adjust `head_capacity` during the entire dataset's execution phase.
> For example, in the case of the Llama-3-8B-Instruct, which consists of 32 layers with 32 heads per layer, the length of the head_score_list is 32×32=1024. For larger models, such as the Llama-3-70B-Instruct, which has 80 layers with 64 heads per layer, the corresponding length of the head_score_list is 80×64=5120. The times required to execute the above code for these models are shown as follows:
>
>
>
> |                      |                        |                       |                        |                        |
> | -------------------- | ---------------------- | --------------------- | ---------------------- | ---------------------- |
> | Model                | Round1 (/s)            | Round2 (/s)           | Round3 (/s)            | Average (/s)           |
> | Llama-3-8B-Instruct  | 0.00032 | 0.00031 | 0.00028 | 0.00030 |
> | Llama-3-70B-Instruct | 0.00135  | 0.00147 | 0.00153  | 0.00145  |
>
>
> The above results indicate that although the initialization time increases with the size of the importance score distribution, the impact remains minimal because:
>
> 1. The time required for this operation is negligible compared to the decoding time (43.8s for FullKV cache to generate 512 tokens).
>
> 2. Initialization only needs to be performed once during the entire runtime, and no dynamic adjustments are required. As the dataset size increases, the initialization time can be further amortized.
>
>
> **Q4: computational efficiency**
>
> For computational efficiency, Figure 6 demonstrates that our KV cache method achieves comparable computational efficiency to other KV cache compression baselines under the same settings. This means our proposed method delivers significant performance improvements without introducing additional overhead.
>
> Regarding the specific results, on Longbench, we analyzed summarization tasks across three datasets: Gov-Report, QMSum, and Multi-News. The average generation length across these datasets is 406.33, with Gov-Report having an average result length of 817.4. These statistics are based on tokenized results using the tokenizer from Llama-3-8B-Instruct. Below, we present a speed comparison between the FullKV cache method and our proposed method when the generation length is set to 512 and 1024 separately:
>
> |   |   |   |   |   |   |
> |---|---|---|---|---|---|
> |Method|Generation length|round1|round2|round3|Average time/s|
> |FullKV|512|47.75|41.75|41.95|43.82|
> |Ours|512|21.06|21.73|30.26|21.01|
> |||||||
> |FullKV|1024|90.68|81.12|81.15|84.32|
> |Ours|1024|42.65|37.70|39.14|39.83|
>
> The KV cache compression method achieves approximately a 2x speedup compared to the original FullKV method (with FlashAttention). We believe this improvement is significant, as FlashAttention itself already provides a substantial speedup (around 4x, as demonstrated in their paper) compared to vanilla attention.
>
> Besides the decoding latency discussed above, another point is Peak Memory Usage. While the difference in memory usage becomes more noticeable when the context length reaches 32k tokens, we believe that any reduction in GPU memory usage is meaningful, as it directly contributes to the efficiency and scalability of large language models. Furthermore, there are now numerous benchmarks exceeding 100k tokens, demonstrating that scenarios involving long contexts are highly relevant. In this work, we followed the settings of previous studies and primarily conducted tests and performance comparisons on LongBench.

---

> ### Author Response · Authors · 2024-11-21
> **Response to Reviewer vGF6 [3/3]**
>
> **Q5: experiment condition and memory usage for context encoding**
>
> For the Peak Memory Usage diagram in Figure 6, we set the generation length to 1. In our paper, we focus on the selection and eviction of KV cache during the pre-filling phase. Therefore, we set the generation length to 1 to emphasize the pre-filling stage. The experimental results show that although KV cache calculations are still required during the pre-filling phase, our proposed method, along with other KV cache compression baselines, can effectively optimize overall GPU memory usage.
>
>
>
>
> **Q6: Fragmented KV caches**
>
> To select KV cache at the head-level and retain different sizes for each head, this may lead to certain discontinuity issues compared to layer-level KV cache compression methods. However, the experimental results in Figure 6 demonstrate that, in sequential execution, head-level KV cache compression methods do not negatively impact overall computational efficiency.
>
> To further analyze the impact of head-level KV cache compression in parallel computation, we conducted experiments comparing the effects of our proposed head-level KV cache method and other KV cache methods, including other head-level methods. These experiments were performed using two A6000 GPUs, with experimental settings and models consistent with those used in the decoding latency experiments in Figure 6. The results are as follows:
>
> |   |   |   |   |   |   |
> |---|---|---|---|---|---|
> |Method|Generation length|round1|round2|round3|Average time/s|
> |FullKV|512|64.02|63.13|63.00|63.38|
> |SnapKV|512|36.57|36.40|36.52|36.50|
> |Ada-SnapKV|512|44.32|45.04|44.84|44.73|
> |Ours|512|46.17|46.25|46.10|46.17|
>
>
> Compared to sequential execution, the head-level method introduces some overhead in parallel computation due to the head-level operations it requires. Therefore, there is a trade-off between performance and speed. As shown in Table 1, head-level KV cache compression methods, such as Ada-SnapKV and our proposed method, achieve better performance compared to layer-level KV cache compression methods like SnapKV and PyramidKV. Moreover, compared to Ada-SnapKV, our method achieves better performance, while maintaining the same computational efficiency.
>
>
>
> **Q7: Compare with RAG methods / other efficiency-concerned papers**
>
> Thank you for your valuable suggestions and the provided related papers. We added one discussion section in our revised version. (Appendix H)
>
> context compression is an interesting direction closely related to KV cache compression. For example, ICAE employs an additional trained In-Context AutoEncoder to compress the input into a fixed-length memory slot, which is then used as the input to the model. From the perspective of ICAE, current KV cache compression methods can be seen as compressing input context using the model's own knowledge. For instance, the SnapKV method, which our approach builds upon, uses the last alpha tokens as the observation window and selects retained KV cache based on attention from these tokens.
>
> Compared to context compression methods like ICAE, current KV cache compression methods are simpler, as they do not require training additional models. They are also easier to obtain higher computational efficiency since these KV cache compression methods avoid the need to rely on external models to obtain compressed inputs.

---

> ### Author Response · Authors · 2024-11-25
> **Follow-up**
>
> Dear Reviewer vGF6;
>
> Thank you again for your valuable input. We believe your suggestions were incredibly insightful, and we have provided detailed responses addressing your concerns, including the motivations and innovations of our paper (Q1), its performance in other scenarios (Q2), the impact of model size on overhead (Q3), computational efficiency (Q4), and the effects of fragmented KV cache (Q5). We would like to know if our responses addressed your concerns and provided satisfactory clarification？
>
> Additionally, we have included further discussions on related methods, particularly regarding context compression, in Appendix H. We are more than willing to engage in further discussions to refine and enhance the paper. Thanks!

---

> ### Comment · Reviewer_vGF6 · 2024-11-25
> **Reviewer's reply**
>
> Q1: Thanks for emphasizing the contribution of this paper. I understand that the two major contributions, but they might be marginal to prior works.
>
> Q2: Thanks for clarifying the scope of this paper. From the writing of the paper, the method seems like a general approach for all kinds of text generation. It would be better if the authors could mention the limitations of their method in the paper.
>
> Q3: Thanks for conducting these analysis. I am not concerned if the scoring or the selection process are slow but whether reducing the KV cache is necessary. [This blob post](https://www.adamcasson.com/posts/transformer-flops) about transformer flop demonstrates that the quadratic terms are only a tiny part of the overall transformer computation, given the majority of the compute is on MLPs, especially for larger models. In another word, the sequence length must be very long to make the KV cache compression to make sense.
>
> Q4, Q5, and Q6: Thanks for providing more details. Given this paper cares about efficiency, it would be great to incorporate them in the next-version of this paper.
>
> Additional discussion about related work: Thanks for doing that.
>
> Overall: Given these clarifications and improvements, I am willing to increase my score to 6. Sorry for replying very late. If authors find it hard to further discuss on any of the above topics, please do them in the paper.

---

> > ### Author Response · Authors · 2024-11-26
> > **Response to Reviewer vGF6**
> >
> > Dear Reviewer vGF6;
> >
> > Thank you for your thoughtful feedback and valuable suggestions. We truly appreciate you revising your score. We will add an additional Limitation section to further clarify the application scenarios and related issues.
> >
> > Regarding the blog you mentioned, We sincerely appreciate the relevant materials you have provided. Current long-context scenario can easily exceed its reported max length, as many modern models now support 128K input lengths, such as Llama-3.1 and Qwen2. Furthermore, there are corresponding ultra-long-length benchmarks like InfiniteBench[1] and RLUER[2], which highlight the importance of the KV cache compression method to some extent. Overall, we are very grateful for your insights and responses.
> >
> > Additionally, we will carefully consider how to integrate Q4, Q5, and Q6 into the revised version of the paper. Thank you once again for your suggestions, which have greatly contributed to making our paper more comprehensive and meaningful.
> >
> > [1] InfiniteBench: Extending Long Context Evaluation Beyond 100K Tokens
> >
> > [2] RULER: What’s the Real Context Size of Your Long-Context Language Models?

---

### Official Review · Reviewer_NjWN · 2024-11-01

**Soundness:** 3
**Presentation:** 2
**Contribution:** 2
**Rating:** 6
**Confidence:** 4

**Summary:**

The paper proposes a new KV cache compression technique, HeadKV and HeadKV-R2, by selectively discarding or retaining the important KV cache based on different types of head. The proposed method consists of two main stages: retrieval-reasoning head identification and KV cache budget allocation based on the head's type. The paper proposes a new type of NIAH test and claim it can be used to identify heads that can both retrieve and reason about the tokens from the context based on the proposed retrieval-reasoning score estimation equation. After successfully identifying the R2-heads, the method dynamically allocate the KV cache budget to retain for each head based on its score from the previous step. The paper claims to maintain 97% performance, compared with baseline with full KV, even though only retaining 1.5% of total KV.

**Strengths:**

- The study presents a new way to compress KV cache based on different types of attention head is novel, even though there are concurrent works that also work on the same idea.
- The performance of the proposed method surpasses other baselines considered in the paper by a considerable margin at the extreme cases when retained KV size is small.

**Weaknesses:**

1. Even though the paper claims to successfully identify heads that can do both retrieval and reasoning for **long-context** task, which is better than the retrieval-only heads initially proposed in Wu et al. (2024), the NIAH experiment setting in the paper is not long enough (longest prompt = 8k). I believe 8k is considered to be not long enough nowadays. Can the author try longer NIAH test such as 64k or 128k to show the effectiveness of the identified R2-heads?
2. I believe there is a work called "Razorattention" [1] that is released in July which follows the same trajectory as this study, i.e. kv cache compression based on head type. Even though the paper addresses this work in their writing, I don't see any comparison in term of performance between the proposed method and that work, especially two works follow the same directory and Razorattention was released a few moths ago. It is unclear to notice the major contribution of the R2-heads from retrieval head only. Can the author benchmark and compare their performance in your experiment?
3. The estimation equation used to determine R2-head seems to be vague (or even incorrect).
- What is the first sigma, or the t-sigma, sum used for?
- The claim that this proposed estimation method can identify which head is responsible for reasoning is not convincing. Firstly, the study modifies the needle to include reasoning logics & incorrect answer, but do not consider them at all in the estimation equation. The estimation equation only considers the correct answer, c2, as the ground truth, which imo, the same as the original retrieval-identification test. What is the usage of the add-on logics and incorrect answer here if you don't consider it in the estimation?
4. I believe the paper would benefit more from ablation study showing and discussing the effect of different values of hyper-parameter alpha & beta on the performance of the methods.

[1] Hanlin Tang, Yang Lin, Jing Lin, Qingsen Han, Shikuan Hong, Yiwu Yao, and Gongyi Wang. Razorattention: Efficient kv cache compression through retrieval heads, 2024. URL https: //arxiv.org/abs/2407.15891.

**Questions:**

Please see weaknesses.

---

> ### Author Response · Authors · 2024-11-21
> **Response to Reviewer NjWN [1/3]**
>
> **Q1:  NIAH experiment setting in the paper is not long enough (longest prompt = 8k)**
>
>    In the NIAH experiments, we followed the settings from SnapKV and PyramidKV and conducted the experiments within the maximum training length supported by each model. Since the maximum supported length for the Llama-3-8B-Instruct model is 8k, the maximum length in Figure 5 is set to 8k. In Figure 8, we present the NIAH experiment results for the Mistral-7B-Instruct model, with a maximum length of 32k. The experimental results on Mistral are consistent with those on Llama-3, which demonstrates the effectiveness of our method in the NIAH task.
>
> **Q2: RazorAttention benchmark**
>
>    Thank you for your suggestion. In Table 1, we also provide the results of head-level KV cache budget allocation based on the standard retrieval head distribution (Head-R).  Since RazorAttention has not published their code and did not provide details on the experimental setup used to obtain their results, we implemented their approach and conducted comparative experiments on Llama-3-8B-Instruct by ourself.
>
>   Following the settings provided in the PyramidKV codebase, we use a window size of 8 and an attention sink size of 4 for reproduction. Taking six QA datasets from Longbench as examples, their average length is 8640. Therefore, to ensure a fair comparison, we need to maintain a consistent number of total KV cache in the model after performing KV cache eviction. When the KV size is 128, we can obtain a total of (128 - 8 - 4) * 32 * 32 = 118,784 KV cache entries for those retrieval heads to maintain a full KV cache. Considering the average length of 8640, the number of retrieval heads that can maintain a full KV cache is 118,784 / 8640 ≈ 14. Therefore, we set the number of retrieval heads (chosen based on the retrieval score) to 5, 10, and 20 for a fair comparison. Results are shown below:
> | Method    | hyper-parameters | NartvQA | Qasper | MF-en | HotpotQA | 2WikiMQA | Musique | Avg   |
> | --------- | ---------------- | ------- | ------ | ----- | -------- | -------- | ------- | ----- |
> | FullKV    | -                | 25.56   | 32.07  | 39.71 | 43.57    | 35.28    | 21.18   | 32.90 |
> | Razor     | 5                | 9.69    | 7.76   | 9.65  | 25.21    | 17.76    | 8.14    | 13.03 |
> | Razor     | 10               | 9.69    | 7.28   | 9.42  | 25.46    | 17.34    | 8.00    | 12.87 |
> | Razor     | 20               | 9.69    | 7.34   | 9.33  | 26.21    | 18.28    | 7.95    | 13.13 |
> | HeadKV-R2 | -                | 21.80   | 29.19  | 41.89 | 43.73    | 35.01    | 20.40   | 32.00 |
>
> When the KV size is 1024,  we can obtain a total of (1024 - 8 - 4) * 32 * 32 = 1,036,288 free KV cache budget and the numbers of Retrieval heads with full KV cache should be: 1,036,288 / 8640 ≈ 120. Therefore, we set the number of retrieval heads (maintaining a full KV cache) to 100 and 500 for a fair comparison:
> | Method    | hyper-parameters | NartvQA | Qasper | MF-en | HotpotQA | 2WikiMQA | Musique | Avg   |
> | --------- | ---------------- | ------- | ------ | ----- | -------- | -------- | ------- | ----- |
> | FullKV    | -                | 25.56   | 32.07  | 39.71 | 43.57    | 35.28    | 21.18   | 32.90 |
> | Razor     | 100              | 10.83   | 8.37   | 11.43 | 27.14    | 19.13    | 11.2    | 14.68 |
> | Razor     | 500              | 24.08   | 31.53  | 38.24 | 36.32    | 29.49    | 17.81   | 29.58 |
> | HeadKV-R2 | -                | 24.66   | 30.82  | 39.56 | 43.97    | 36.47    | 22.24   | 32.95 |
>
> The results indicate that RazorAttention heavily relies on setting a large number of retrieval heads to maintain full KV cache in order to achieve good performance. This suggests that simply using streamingLLM on non-retrieval heads is insufficient for effectively retaining important information. Under the same KV cache settings, our method significantly outperforms RazorAttention, as we dynamically allocate KV cache size based on importance scores and use SnapKV to select the retained KV cache for each head.
>
>
> Another noteworthy point when reproducing RazorAttention is that its method of compressing information from dropped tokens into a “compensation token” is highly time-consuming. When only the top k=1000 retrieval heads maintain full KV cache, the time required to run NartvQA is three times longer than with top k=5 retrieval heads (16 minutes vs. 5 minutes), which contradicts the goal of KV cache compression.

---

> ### Author Response · Authors · 2024-11-21
> **Response to Reviewer NjWN [2/3]**
>
> **Q3.1:  estimation equation used to determine R2-head**
>
> We refine the estimation method by focusing on the entire answers $c^2$ rather than the only token with the highest attention probability. Eq. 2 can be used to obtain the importance score for each head. For this equation:
> 1. (The i-sigma): We consider the tokens with the top-i highest probabilities and compute the importance score by accumulating the attention scores of the tokens that appear in the correct answer. This basic motivation here is: first, if one head is important, it can pay attention to all the tokens within the correct answer (why top-$i$). Second, tokens with a higher attention score should contribute more to the importance score if these tokens can be found inside of the correct answer (why use attention score $a_i$).
> 2. (The t-sigma):  We follow the setup of Eq. 1 to accumulate the importance score for each head step-by-step.
> Ideally, The maximum score for the i-sigma (inner sum) should be 1/N. By accumulating the importance score step-by-step (the t-sigma), the maximum score for each head should also be 1, which is the same as Retrieval Heads. Importance score distributions shown in Figure 4  are normalized distributions. Compared to standard Retrieval Heads Distribution, our new distribution should be more dense, since we focus on the total correct answer rather than only focusing on the token with the highest attention score. A denser distribution plays an important role in guiding KV cache eviction, as it allows us to set the KV cache size more specifically for each head. This is something that the standard retrieval head distribution cannot achieve, as around 70% of heads receive a value of zero in the standard retrieval head distribution, making it impossible to effectively distinguish and allocate budget for these heads.
>
>
> **Q3.2:  add-on logics and incorrect answer.**
>
> The addition of reasoning logics and incorrect answers aims to introduce complexity and context to the reasoning process, which we believe might highlight different heads depending on whether the head supports accurate reasoning patterns or not. Here’s how we envisioned their role:
>
> + *Aligning with the requirements of contextual reasoning:* Based on the in-depth analysis on the contextual reasoning dataset, we know that the answer to the corresponding question will still appear in the input but with various distractors. Therefore, the model continues to rely on the retrieval-and-paste mechanism to obtain the true answer. The original retrieval heads estimation method did not account for this phenomenon, but we address it by adding logic and simulating incorrect answers to achieve a more accurate distribution.
>
> + *Introducing diverse reasoning paths:* By incorporating both reasoning content and incorrect answers, we are simulating two different potential reasoning paths. The incorrect answer acts as a distractor and we hope to find heads that concentrate on the correct answer even though the incorrect answer has almost the same structure with the correct answer. We expected the correct answer $c^2$ to be treated as the ground truth in the estimation equation, similar to the original Retrieval Heads estimation method.
>
> + *Focusing on the correct reasoning path:* The purpose of constructing retrieval-reasoning examples is to obtain the importance score for each head, which then guides the head-level KV cache budget allocation. Therefore, our goal is to identify the important heads rather than those focused on incorrect answers. By emphasizing the important heads, the heads that focus on incorrect answers are naturally ignored, as they share the same shared global budget pool.
>
> + *Aligning with the standard retrieval heads estimation:* we followed the setup in obtaining the Retrieval Heads and determined the Retrieval-Reasoning Heads distributions  based on the NIAH experiments. Since NIAH only outputs the corresponding results, we choose to focus on the correct answer to ensure that the maximum importance score each head can achieve in Eq. 2 is 1. Adding additional logic would disrupt this property, potentially affecting the final distribution.
>
> We agree that retrieval heads play an important role in guiding head-level KV cache allocation, as shown by the Head-R results in Table 1, which also significantly outperform the other baselines. However, due to the sparsity issue, they are less effective in guiding head-level KV cache budget allocation. To address this, we: (1) incorporated retrieval-reasoning examples, and (2) refined the importance score estimation. The results in Table 2 demonstrate that using only retrieval examples as defined in retrieval heads (Head-R) or plus refining the importance score estimation method (Head-ER) does not achieve optimal performance. Therefore, both constructing retrieval-reasoning examples and refining the importance score estimation are necessary for optimal results (Head-R2).

---

> ### Author Response · Authors · 2024-11-21
> **Response to Reviewer NjWN [3/3]**
>
> **Q4:the effect of different values of hyper-parameter**
>
> We included one section about hyper-parameter $\beta$ into our revised version in Appendix G. The only hyper-parameter introduced by our method is $\beta$, which defines the size of the shared global budget pool $B$. Other hyper-parameters, such as the number of instruction tokens $\alpha$, are kept consistent with the settings provided in the PyramidKV codebase. We also ensure that all other hyper-parameters are consistent across both the baselines and our proposed method. For the hyper-parameter $\beta$, as we said in Section 4.1 Line 305, it was chosen from {1.005, 1.01, 1.1, 1.2, 1.5, 2, 5, 10} and we report the best performance.
>
> For $\beta$, a smaller value represents a larger shared budget pool $B$, meaning that KV cache allocation relies more heavily on the importance score distribution for allocation. The results in appendix G show that Head-R2 performs better with a smaller $\beta$, indicating that our retrieval-reasoning head distribution is more effective in guiding KV cache budget allocation.

---

> > ### Comment · Reviewer_NjWN · 2024-11-22
> >
> > Thank the authors for providing a detailed rebuttal, and I'm satisfied with it. I have increased my score.

---

> > > ### Author Response · Authors · 2024-11-22
> > > **Response to Reviewer NjWN**
> > >
> > > We sincerely appreciate the time and effort you have dedicated to reviewing out work. Your valuable feedback and constructive suggestions have been instrumental in improving the quality of our research. Thank you for your thoughtful insights!

---

### Official Review · Reviewer_NPoe · 2024-11-04

**Soundness:** 4
**Presentation:** 3
**Contribution:** 3
**Rating:** 6
**Confidence:** 4

**Summary:**

This paper proposes a head-level KV cache eviction method, and the authors use the retrieval capability of heads to score their importance, then less important heads evict more tokens. Based on previous retrieval-head criteria, the authors put forward two improvements: one is a more challenging retrieve-reasoning dataset, and the other is using attention weights to refine the score, making the scoring more accurate. By setting different KV cache budges for different heads, this method outperforms other baselines in various evaluation metrics while maintaining the same total KV cache size and inference latency.

**Strengths:**

1. The KV cache budget allocation strategy maintains the total amount of KV cache constant and keeps inference time unchanged.

2. Using top-k attention weights to refine the score can enhance the retrieval-head evaluation.

3. The proposed retrieve-reasoning dataset may benefit future works.

**Weaknesses:**

1. How the S_h is normalized is not mentioned in this paper. In Equation (4), it seems that S_h should sum to one across all heads and all layers.

2. How the retrieval-reasoning dataset is generated is not mentioned in the paper.

3. The left subfigure in Figure 6 saids decoding times but line 512 mentioned that the decoding time includes prefilling time. This is quite confusing. In the figure, the prefill time of each method (when generating a length of 0) is squeezed into the same point on the image, making it impossible to discern their merits and demerits. Please rename the axis label, move the relevant explanation to the caption, or separate the prefill and decode into two figures.

**Questions:**

1. The a_h in Equation (1) and (2) may need a superscript t.

2. Typo in Figure 1: The red text Prefilling Phrase should be Prefilling Phase.

3. Typo in Figure 5 and Figure 8: 'Neele-in-a-Haystack' should be 'Needle'.

4. Are those gaps between FullKV and HeadKV in the right subfigure of Figure 6 solely caused by KV cache?

5. Giving a pseudo code would be better.

---

> ### Author Response · Authors · 2024-11-21
> **Response to Reviewer NPoe**
>
> **Q1: S_h normalization**
>
> Thanks for pointing out that problem. S_h​ is L1-normalized after collecting the importance score distribution. We ensure that the sum of S_h​ equals 1 to guide the subsequent head-level KV cache budget allocation.
>
> **Q2: How to construct retrieval-reasoning dataset**
>
> Thanks for pointing out, We included the details on how to construct the retrieval-reasoning dataset in the appendix E of the revised version.
>
> For guiding head-level KV cache compression, we need to obtain the importance score for each head. To achieve this, we manually construct specific examples to ensure that the model relies on heads rather than internal knowledge to answer questions during the Needle-in-a-Haystack experiment. Therefore, we construct retrieval-reasoning examples based on retrieval examples by introducing different reasoning paths into examples to emphasize the contextual reasoning ability. One constructed retrieval-reasoning example is shown in Figure 2. In addition to the current example, we reverse the question to create a total of two examples for the Needle-in-a-Haystack experiment.
>
> Following the setup outlined in the Retrieval Heads paper, in the Needle-in-a-Haystack experiment, we use the model's maximum training length as the maximum haystack length and evenly select 5 different length values as the actual haystack lengths. For each haystack length, the question is inserted at 10 different depths, uniformly distributed from the start to the end of the current haystack length. In total, we generate 100 examples per model to collect Retrieval-Reasoning Head distributions.
>
>
> **Q3: Misleading between decoding latency and decoding time**
>
> Thanks for your feedback. We fixed those errors in the revised version. We should change the y-axis in the left subfigure of Figure 6 to "time". The description in Line 512 is correct—decoding latency includes both the pre-filling time and the decoding time. Pre-filling refers to the KV cache eviction phase performed after each example is encoded by the model, while decoding refers to the generation of the output after the pre-filling phase is completed. Therefore, when the generation length is set to 1, the decoding latency reflects the time required for the model to encode the current input and perform pre-filling. The detailed decoding latency results when the generation length is set to 1 for our strong baseline method Ada-SnapKV and our method are shown below:
> | Method     | round1 | round2 | round3 | Average time/s |
> | ---------- | ------ | ------ | ------ | -------------- |
> | FullKV     | 4.25   | 4.70   | 4.34   | 4.43           |
> | Ada-SnapKV | 4.58   | 5.50   | 5.41   | 5.16           |
> | Ours       | 4.42   | 5.01   | 4.66   | 4.69           |
>
> Based on the average results from three rounds, our method does not introduce significant additional time. In contrast, Ada-SnapKV may require extra time to compute attention and perform sorting to determine the corresponding allocation strategy. Our Retrieval-Reasoning Head distribution is a static distribution, allowing the allocation strategy to be obtained with minimal overhead. However, the pre-filling time is still negligible compared to the decoding time when generation length is relatively large.
>
> **Q4: Typo Error & pseudo code**
>
> Thanks for your careful reading, and we will fix those errors in the new version. We also add a pseudo code in Appendix F. Our code is based on the PyramidKV and Ada-KV implementations, and we will make our code publicly available. Compared to Ada-SnapKV, our main difference lies in constructing the corresponding allocation strategy for each head. Since the obtained standard Retrieval Heads distribution and Retrieval-Reasoning Heads distribution are static and do not change with modifications to the input, we can perform a one-time initialization when the model is loaded.
>
> **Q5: gaps between FullKV and HeadKV solely caused by KV cache**
>
> In the Peak Memory Usage results shown in Figure 6, we compared memory usage while maintaining consistency in the input context length. Unlike FullKV, all other KV cache compression methods limit their modifications to the selective eviction of KV cache. Therefore, we can conclude that the memory gap between FullKV and the other KV cache compression methods is entirely caused by differences in KV cache related operation. We attempted to further trace different tensors and analyze the tensors stored in GPU memory. However, we were only able to identify tensors corresponding to model weights, inputs, and RoPE, which remain constant between FullKV and all other KV cache compression methods. We could not directly trace tensors related to the KV cache. This limitation is likely due to the use of additional variables and data structures within the transformers framework to manage KV cache storage.

---

> > ### Author Response · Authors · 2024-11-25
> > **Follow-up**
> >
> > Dear Reviewer NPoe;
> >
> > We greatly appreciate your careful reading and thoughtful feedback. Based on your suggestions, we have added Appendix E to explain how the dataset was constructed and Appendix F to include pseudo code, providing a clearer demonstration of our proposed method. These additions have indeed made our paper more comprehensive and complete. Please let us know if our responses fully addressed your concerns. We are always open to further discussion and welcome any suggestions to further improve our work!

---

> > ### Comment · Reviewer_NPoe · 2024-11-25
> >
> > Thank you for your response. It has addressed my concern. Since my initial score is already positive, I will maintain my current score.

---

### Official Review · Reviewer_iyGe · 2024-11-05

**Soundness:** 4
**Presentation:** 4
**Contribution:** 3
**Rating:** 8
**Confidence:** 3

**Summary:**

This paper proposed a head-level Key-Value cache compression algorithm, different from Ada-KV (also head-level), they don’t perform  allocation within a single layer during the budget allocation process but for all heads.

They conduct experiments on LongBench and LooGLE, the performance is consistently better or comparable than the full KV baseline with a reasonable amount of KV cache size.

Also, they introduced the the retrieval-reasoning head to assign higher importance score for those heads with higher attentions on the correct answer. Such head seems useful based on the experiments, though not all of them, still makes reasonable improvements on certain datasets.

They also conduct thorough analysis on long-context retrieval, latency , memory usage, etc.

**Strengths:**

1. Reasoning head-level kV cache allocation and importance score estimation.
2. Performance can be consistently better or comparable with the full KV setting.

**Weaknesses:**

1. We do not see much improvement on latency and memory as compared against Ada-KV, as I believe this work is based on Ada-KV.

**Questions:**

N.A.

---

> ### Author Response · Authors · 2024-11-21
> **Response to Reviewer iyGe**
>
> **Q1: Computational Efficiency compared to Ada-KV**
>
> Thank you for your summary and it is very accurate and comprehensive. While maintaining a consistent KV cache, our method significantly outperforms other baselines, including the Ada-KV method. This indicates that we can achieve results comparable to Ada-SnapKV and other baselines while using a smaller KV cache. For example, in Table 4, on the Llama-3-8B-Instruct model, our proposed method HeadKV-R2 achieves a score of 32.51 with a KV size of 256, surpassing Ada-SnapKV’s score of 32.31 with a KV size of 1024. The same conclusion holds for the Mistral-7B-Instruct model, where HeadKV-R2 achieves 32.24 with a KV size of 256, compared to Ada-SnapKV’s 31.98 with a KV size of 1024.

---

### Public Comment · ~Sophia_Fulton1 · 2024-11-15
**Confusion Regarding Hyper-parameter $\beta$ Selection**

Hi, authors. While reading this paper, I found the statement in the Experiment Settings section—"The hyper-parameter $\beta$, which controls the size of the shared budget pool, was chosen from {1.005, 1.01, 1.1, 1.2, 1.5, 2, 5, 10}, and we report the best performance"—somewhat unclear. Does this mean that HeadKV used different $\beta$ values for various budgets or datasets to achieve the reported results?

It would be helpful if the paper could specify the exact $\beta$ values corresponding to the results presented, as this would improve the transparency of the methodology. Additionally, providing guidelines on how to select $\beta$ without extensive searching would be valuable, as this approach seems impractical in real-world applications.

---

> ### Author Response · Authors · 2024-11-21
> **Response to Sophia Fulton**
>
> We really appreciate your attention!
>
> In the revised version, we have added an analysis of the hyperparameter $\beta$ in Appendix G. Conducting a grid search to identify the optimal hyperparameter is a common practice in NLP research. In our case, we maintained consistency within the benchmark datasets, meaning we did not fine-tune $\beta$ for each individual dataset to obtain the best possible results. While such dataset-specific tuning could lead to better overall results, it would deviate from the original intent of our paper. As shown in the analysis in Appendix G, our proposed method consistently outperforms the best baseline, Ada-SnapKV, under various $\beta$ settings. This demonstrates the robustness of our approach.
>
> Regarding the selection of $\beta$, we recommend using a smaller value as an initialization and use a small calibration dataset to determine the $\beta$. This aligns with the experimental results in Appendix G, where smaller $\beta$ values yield better performance. Additionally, a smaller $\beta$ indicates that the model relies more on the provided importance score distributions to allocate budget. When $\beta$ is set to an extremely large value, causing the shared budget pool $B$ to reduce to zero, our algorithm degenerates into SnapKV. Thus, SnapKV can be considered the lowest bound of our proposed method.

---

> > ### Public Comment · ~Sophia_Fulton1 · 2024-11-22
> > **Appreciation and Suggestions for Explicit Hyperparameter $\beta$ Settings**
> >
> > Thanks for your response. The ablation study presented in the appendix indeed highlights the excellent performance of HeadKV-R2. However, I am a bit puzzled by one observation: it appears that smaller values of $\beta$ consistently lead to better performance. This raises the question of why a grid search is necessary in the evaluation.
> >
> > Additionally, I have concerns regarding the hyperparameter $\beta$ tuning process, mainly due to the risk of data leakage. Typically, grid searches are conducted on the training set or a validation set, which is strictly separated from the final test set to prevent such risks. However, the paper does not provide details on whether the training or validation set was kept separate from the test set in evaluation.
> >
> > I would strongly encourage the authors to specify the detailed $\beta$ settings, such as the values used for different models and budgets. This would be immensely valuable for researchers aiming to follow up on this work, as it could significantly reduce the cost of conducting extensive grid searches to reproduce these results.

---

> > > ### Author Response · Authors · 2024-11-25
> > > **Response to Sophia Fulton - 2**
> > >
> > > Thanks for your valuable feedback! We have uploaded a revised version, providing the optimal $\beta$ settings corresponding to the results in Appendix Table 4. Additionally, we also add the results of selecting $\beta$ using a validation set under different KV size settings on Llama3-8B in Figure 11.
> > >
> > > Since LongBench does not provide a validation set, we used scikit-learn to randomly split datasets from LongBench into validation and test sets and reported the corresponding results. The outcomes shown in Figure 11 are well-aligned with the settings presented in Table 4. It is also worth noting that when the KV size is set to 1024, adjustments to $\beta$ within a certain range ([1.005, 1.01, 1.1, 1.2]) have minimal impact on the results, further demonstrating the robustness of our proposed method with respect to the hyperparameter $\beta$. Moreover, as shown in the new results added in Appendix G, our method significantly outperforms Ada-SnapKV across different $\beta$ values.
> > >
> > > Another point we want to emphasize is that other KV cache compression baselines also incorporate hyperparameters. For instance, the strongest baseline, Ada-SnapKV, utilizes the hyperparameter `floor_alpha` to define the minimum number of KV caches for each head, as outlined in [1], which is quite similar to our proposed $\beta$. Since LongBench does not provide a validation set and Ada-SnapKV does not explain the hyperparameter selection in its paper or code, we conducted necessary tests to ensure a fair comparison in our paper.
> > >
> > > [1] https://github.com/FFY0/AdaKV/blob/main/experiments/LongBench/pred.py#L185

---

> > > > ### Public Comment · ~Sophia_Fulton1 · 2024-12-02
> > > > **Thanks to the Authors for the Clarification**
> > > >
> > > > Thank you for the clarification. It is very helpful to me and will be also valuable for future researchers looking to build on this work.

---

### Author Response · Authors · 2024-11-21
**Summary of our revisions**

**We sincerely thank the reviewers for their thorough reading and valuable feedback.**

**Contribution of the paper:**

Our paper aims to enhance computational efficiency in long-context scenarios. By designing a fully head-level KV cache compression method during the pre-filling stage, we can retain only 1.5% of the KV cache while maintaining 97% of the performance of the full KV cache. To achieve the fully head-level KV cache compression, we first obtain the Retrieval-Reasoning distribution by refining the importance score estimation and constructing the retrieval-reasoning examples to serve as the dataset for estimation. Secondly, We design a head-level KV cache allocation strategy by creating a shared budget pool guided by the importance score distribution to determine the KV cache retention for each head. We believe that head-level KV cache compression is more meaningful and promising, as the significance of different heads can vary considerably, even within the same layer.

**Overview of Revision:**
- We have made revisions to the expressions and formulas based on the reviewers’ feedback.

- We used different colors for the added sections to indicate which reviewer’s comments they correspond to.

- We added a section of how to construct the retrieval-reasoning dataset, along with corresponding explanations and final dataset. (Appendix E)

- We included additional pseudo-code to clarify the settings and overhead of our proposed method. (Appendix F)

- We added an analysis of the hyperparameter $\beta$. (Appendix G)

- We added a discussion section to show the differences between context compression and KV cache compression. (Appendix H)

---

### Public Comment · ~Xiang_Liu10 · 2024-12-03
**Missing Reference to Previous KV Cache Compression Work Based on Attention Head Level**

Hi, authors. While reading this paper, I found you should consider referencing the paper "Model Tells You What to Discard: Adaptive KV Cache Compression for LLMs" (ICLR 2024, Oral), which presents a related approach to KV cache compression based on attention head-level granularity for large language models (LLMs). These two papers share similar motivations, making it relevant to highlight this prior work.

Engineering-Related Question:

If each attention head retains a different number of tokens, how is attention computed? Typically, attention mechanisms assume that each head operates over the same number of tokens. I would appreciate clarification on how this issue is addressed in the current framework.

---

> ### Author Response · Authors · 2024-12-03
> **Response to Xiang Liu**
>
> Thank you for your interest and for pointing out this issue!
>  We will include the relevant citations and highlight this prior work in future version. Our code is based on AdaKV [1], with the distinction that we use retrieval-reasoning heads to guide the head-level KV cache budget allocation. For the implementation, the `flash_attn_varlen_func` function provided in flash-attn[2] handles variable-length sequences for different heads to address the issue of unequal lengths. More settings and implementation details can be found in the code published with the AdaKV [3].
>
> [1] Ada-KV: Optimizing KV Cache Eviction by Adaptive Budget Allocation for Efficient LLM Inference
>
> [2] https://github.com/Dao-AILab/flash-attention/blob/c4b9015d74bd9f638c6fd574482accf4bbbd4197/flash_attn/flash_attn_interface.py#L1051
>
> [3]https://github.com/FFY0/AdaKV

---

### Meta-Review · Area_Chair_8Zcq · 2024-12-20

**Metareview:**

This paper introduces a new key-value (KV) cache compression method that allocates KV cache across attention heads based on their importance. It also presents a variant that refines the importance score considering retrieval and reasoning capabilities, i.e., identifying heads with higher attention to correct answers using a retrieval-reasoning dataset. Results on long-context LM benchmarks and long-context QA demonstrate consistent improvements under varying KV cache budgets.

Strengths
- A new KV cache allocation and importance score estimation (iyGe, NPoe, NjWN)
- Performance is consistently better (iyGe, NjWN)
- The new retrieve-reasoning database may be beneficial for future work (NPoe)

Weaknesses
- Improvement is marginal compared to Ada-SnapKV (iyGe) - In AC’s opinion, improvements are significant with KV size 128 but less with KV size 1024, partially because when KV size = 1024 all KV cache methods achieve performance that is close to full KV.
- Limited novelty (vGF6)
- Limited application scenario – this would be more useful for understanding tasks, summarization tasks, or tasks with CoT where substantial retrieval of the given input text is required, but less useful for other tasks such as long-term text generation (vGF6).

**Additional Comments On Reviewer Discussion:**

Weaknesses address during the rebuttal
- Some details in method need clarification (NPoe, NjWN) -> clarified and acknowledged during rebuttal
- Datasets in the paper are not long enough (8K context window) (NjWN) -> this was unavoidable because llama was trained with 8K context window
- Comparison to RazorAttention (NjWN) -> additional results provided during rebuttal

---

### Decision · Program_Chairs · 2025-01-22

Accept (Poster)